# Effects of $NO_x$ and $SO_2$ on the Secondary Organic Aerosol Formation from Photooxidation of α-pinene and Limonene

Defeng Zhao[1], Sebastian H. Schmitt[1], Mingjin Wang[1, 2], Ismail-Hakki Acir[1, a], Ralf Tillmann[1], Zhaofeng Tan[1, 2], Anna Novelli[1], Hendrik Fuchs[1], Iida Pullinen[1, b], Robert Wegener[1], Franz Rohrer[1], Jürgen Wildt[1], Astrid Kiendler-Scharr[1], Andreas Wahner[1], Thomas F. Mentel[1]

[1] Institute of Energy and Climate Research, IEK-8: Troposphere, Forschungszentrum Jülich, Jülich, 52425, Germany

[2] College of Environmental Science and Engineering, Peking University, Beijing, 100871, China

[a]Now at: Institute of Nutrition and Food Sciences, University of Bonn, Bonn, 53115, Germany; [b]Now at: Department of Applied Physics, University of Eastern Finland, Kuopio, 7021, Finland.

*Correspondence to: Th. F. Mentel (t.mentel@fz-juelich.de)*

## Abstract

Anthropogenic emissions such as $NO_x$ and $SO_2$ influence the biogenic secondary organic aerosol (SOA) formation, but detailed mechanisms and effects are still elusive. We studied the effects of $NO_x$ and $SO_2$ on the SOA formation from the photooxidation of α-pinene and limonene at ambient relevant $NO_x$ and $SO_2$ concentrations ($NO_x$: < 1 ppb to 20 ppb, $SO_2$: <0.05 ppb to 15 ppb). In these experiments, monoterpene oxidation was dominated by OH oxidation. We found that $SO_2$ induced nucleation and enhanced SOA mass formation. $NO_x$ strongly suppressed not only new particle formation but also SOA mass yield. However, in the presence of $SO_2$ which induced high number concentration of particles after oxidation to $H_2SO_4$, the suppression of the mass yield of SOA by $NO_x$ was completely or partly compensated. This indicates that the suppression of SOA yield by $NO_x$ was largely due to the suppressed new particle formation, leading to a lack of particle surface for the organics to condense on and thus a significant influence of vapor wall loss on SOA mass yield. By compensating for the suppressing effect on nucleation of $NO_x$, $SO_2$ also compensated for the suppressing effect on SOA yield. Aerosol mass spectrometer data show that increasing $NO_x$ enhanced nitrate formation. The majority of the nitrate was organic nitrate (57%-77%), even in low $NO_x$ conditions (<~1 ppb). Organic nitrate contributed 7%-26% of total organics assuming a molecular weight of 200 g/mol. SOA from α-pinene photooxidation at high $NO_x$ had generally lower hydrogen to carbon ratio (H/C), compared to low $NO_x$. The $NO_x$ dependence of the chemical composition can be attributed to the $NO_x$ dependence of the branching ratio of the $RO_2$ loss reactions, leading to lower fraction of organic hydroperoxides and higher fractions of organic nitrates at high $NO_x$. While $NO_x$ suppressed new particle formation and SOA mass formation, $SO_2$ can compensate for such effects, and the combining effect of $SO_2$ and $NO_x$ may have important influence on SOA formation affected by interactions of biogenic volatile organic compounds (VOC) with anthropogenic emissions.

## 1 Introduction

Secondary organic aerosol (SOA) have significant impacts on air quality, human health and climate change (Hallquist et al., 2009; Kanakidou et al., 2005; Jimenez et al., 2009; Zhang et al., 2011). SOA mainly originates from biogenic volatile organic compounds (VOC) emitted by terrestrial vegetation (Hallquist et al., 2009). Once emitted into the atmosphere, biogenic VOC can undergo reactions with atmospheric oxidants including OH, $O_3$ and $NO_3$, and form SOA. When an air mass enriched in biogenic VOC is transported over an area with substantial anthropogenic emissions or vice versa, the reaction behavior of VOC and SOA formation can be altered due to the interactions of biogenic VOC with anthropogenic emissions such as $NO_x$, $SO_2$, anthropogenic aerosol and anthropogenic VOC. A number of field studies have highlighted the role of the anthropogenic-biogenic interactions in SOA formation (de Gouw et al., 2005; Goldstein et al., 2009; Hoyle et al., 2011; Worton et al., 2011; Glasius et al., 2011; Xu et al., 2015a; Shilling et al., 2012), which can induce an "anthropogenic enhancement" effect on SOA formation.

Among biogenic VOC, monoterpenes are important contributors to biogenic SOA due to their high emission rates, high reactivity, and relative high SOA yield compared to isoprene (Guenther et al., 1995; Guenther et al., 2012; Chung and Seinfeld, 2002; Pandis et al., 1991; Griffin et al., 1999; Hoffmann et al., 1997; Zhao et al., 2015b; Carlton et al., 2009). The anthropogenic modulation of the SOA formation from monoterpene can have important impacts on regional and global biogenic SOA budget (Spracklen et al., 2011). The influence of various anthropogenic pollutants on SOA formation of monoterpene have been investigated by a number of laboratory studies (Sarrafzadeh et al., 2016; Zhao et al., 2016; Flores et al., 2014; Emanuelsson et al., 2013; Eddingsaas et al., 2012a; Offenberg et al., 2009; Kleindienst et al., 2006; Presto et al., 2005; Ng et al., 2007; Zhang et al., 1992; Pandis et al., 1991; Draper et al., 2015; Han et al., 2016). In particular, $NO_x$ and $SO_2$ have been shown to affect SOA formation from monoterpene.

$NO_x$ changes the fate of $RO_2$ radical formed in VOC oxidation and therefore can change reaction product distribution and aerosol formation. At low $NO_x$, $RO_2$ mainly react with $HO_2$, forming organic hydroperoxides. At high $NO_x$, $RO_2$ mainly react with NO, forming organic nitrate (Hallquist et al., 2009; Ziemann and Atkinson, 2012; Finlayson-Pitts and Pitts Jr., 1999). Some studies found that the SOA yield from α-pinene is higher at lower $NO_x$ concentration for ozonolysis (Presto et al., 2005) and photooxidation (Ng et al., 2007; Eddingsaas et al., 2012a; Han et al., 2016; Stirnweis et al., 2017). The decrease of SOA yield with increasing $NO_x$ was proposed to be due to the formation of more volatile products like organic nitrate under high $NO_x$ conditions (Presto et al., 2005). In contrast, a recent study found that the suppressing effect of $NO_x$ is in large part attributed to the effect of $NO_x$ on OH concentration for the SOA from β-pinene oxidation, and after eliminating the effect of $NO_x$ on OH concentration, SOA yield only varies by 20-30% (Sarrafzadeh et al., 2016). Beside the effect of $NO_x$ on SOA yield, $NO_x$ has been found to suppress the new particle formation from VOC directly emitted by Mediterranean trees (mainly monoterpenes) (Wildt et al., 2014) and β-pinene (Sarrafzadeh et al., 2016), thereby reducing condensational sink present during high $NO_x$ experiments.

Regarding the effect of $SO_2$, the SOA yield of α-pinene photooxidation was found to increase with $SO_2$ concentration at high $NO_x$ concentrations ($SO_2$: 0-252 ppb, $NO_x$: 242-543 ppb, α-pinene: 178-255 ppb) (Kleindienst et al., 2006) and the increase is attributed to the formation of $H_2SO_4$ acidic aerosol. Acidity of seed aerosol was also found to enhance particle yield of α-pinene at high $NO_x$ (Offenberg et al. (2009): $NO_x$ 100-120

ppb, α-pinene 69-160 ppb; Han et al. (2016): initial NO ~70 ppb, α-pinene 14-18 ppb). In constrast, Eddingsaas et
al. (2012a) found that particle yield increases with aerosol acidity only in "high NO" condition ($NO_x$ 800 ppb, α-
pinene: 20-52 ppb), but is independent of the presence of seed aerosol or aerosol acidity in both "high $NO_2$"
condition ($NO_x$ 800 ppb)" and low $NO_x$ ($NO_x$ lower than the detection limit of the $NO_x$ analyzer). Similarly, at
low $NO_x$ (initial NO <0.3 ppb, α-pinene ~20 ppb), Han et al. (2016) found that the acidity of seed has no
significant effect on SOA yield from α-pinene photooxidation. In addition, $SO_2$ was found to influence the gas
phase oxidation products from α-pinene and β-pinene photooxidation, which is possibly due to the change in
$OH/HO_2$ ratio caused by $SO_2$ oxidation or $SO_3$ directly reacting with organic molecules (Friedman et al., 2016).
While these studies have provided valuable insights into the effects of $NO_x$ and $SO_2$ on SOA formation, a
number of questions still remain elusive. For example, many studies used very high $NO_x$ and $SO_2$ concentrations
(up to several hundreds of ppb). High $NO_x$ can make the $RO_2$ radical fate dominated by one single pathway (i.e.,
$RO_2$+NO or $RO_2$+$NO_2$) to investigate SOA yields and composition under such conditions. Yet, the effects of $NO_x$
and $SO_2$ at concentration ranges for ambient anthropogenic-biogenic interactions (sub ppb to several tens of ppb
for $NO_2$ and $SO_2$) have seldom been directly addressed. Moreover, many previous studies on the SOA formation
from monoterpene oxidation focus on ozonolysis or do not distinguish OH oxidation and ozonolysis in
photooxidation, and only a few studies on OH oxidation have been conducted (Eddingsaas et al., 2012a; Zhao et
al., 2015b; McVay et al., 2016; Sarrafzadeh et al., 2016; Henry et al., 2012; Ng et al., 2007). More importantly,
studies that investigated the combined effects of $NO_x$ and $SO_2$ are scarce, although they are often co-emitted from
anthropogenic sources. According to previous studies, $NO_x$ can have a suppressing effect on SOA formation
while $SO_2$ can have an enhancing effect. $NO_x$ and $SO_2$ might have counteracting or synergistic effects in SOA
formation in the ambient atmosphere.
In this study, we investigated the effects of $NO_x$, $SO_2$ and their combining effects on SOA formation from the
photooxidation of α-pinene and limonene. α-pinene and limonene are two important monoterpenes with high
emission rates among monoterpenes (Guenther et al., 2012). OH oxidation dominated over ozonolysis in the
monoterpene oxidation in this study as determined by measured OH and $O_3$ concentrations. The relative
contributions of $RO_2$ loss reactions at low $NO_x$ and high $NO_x$ were quantified using measured $HO_2$, $RO_2$, and NO
concentrations. The effects on new particle formation, SOA yield, and aerosol chemical composition were
examined. We used ambient relevant $NO_x$ and $SO_2$ concentrations so that the results can shed lights on the
mechanisms of interactions of biogenic VOC with anthropogenic emissions in the real atmosphere.
**2  Experimental**
**2.1      Experimental setup and instrumentation**
The experiments were performed in the SAPHIR chamber (Simulation of Atmospheric PHotochemistry In a
large Reaction chamber) at Forschungszentrum Jülich, Germany. The details of the chamber have been described
before (Rohrer et al., 2005; Zhao et al., 2015a; Zhao et al., 2015b). Briefly, it is a 270 m³ Teflon chamber using
natural sunlight for illumination. It is equipped with a louvre system to switch between light and dark conditions.
The physical parameters for chamber running such as temperature and relative humidity were recorded. The solar

irradiation was characterized and the photolysis frequency was derived (Bohn et al., 2005; Bohn and Zilken, 2005).

Gas and particle phase species were characterized using various instruments. OH, $HO_2$ and $RO_2$ concentrations were measured using a laser induced fluorescence (LIF) system with details described by Fuchs et al. (2012). OH was formed via HONO photolysis, which was produced from a photolytic process on the Teflon chamber wall (Rohrer et al., 2005). From OH concentration, OH dose, the integral of OH concentration over time, was calculated in order to better compare experiments with different OH levels. For example, experiments at high $NO_x$ in this study generally had higher OH concentrations due to the faster OH production by recycling of $HO_2\bullet$ and $RO_2\bullet$ to OH. The VOC were characterized using a Proton Transfer Reaction Time-of-Flight Mass Spectrometer (PTR-ToF-MS) and Gas Chromatography-Mass spectrometer (GC-MS). $NO_x$, $O_3$ and $SO_2$ concentrations were characterized using a $NO_x$ analyzer (ECO PHYSICS TR480), an $O_3$ analyzer (ANSYCO, model O341M), and an $SO_2$ analyzer (Thermo Systems 43i), respectively. $O_3$ was formed in photochemical reactions since $NO_x$, even in trace amount (<~1 ppbV), was present in this study. More details of these instrumentation are described before (Zhao et al., 2015b).

The number and size distribution of particles were measured using a condensation particle counter (CPC, TSI, model 3786) and a scanning mobility particle sizer (SMPS, TSI, DMA 3081/CPC 3785). From particle number measurement, the nucleation rate ($J_{2.5}$) was derived from the number concentration of particles larger than 2.5 nm as measured by CPC. Particle chemical composition was measured using a High-Resolution Time-of-Flight Aerosol Mass Spectrometer (HR-ToF-AMS, Aerodyne Research Inc.). From the AMS data, oxygen to carbon ratio (O/C), hydrogen to carbon ratio (H/C), and nitrogen to carbon ratio (N/C) were derived using a method derived in the literature (Aiken et al., 2007; Aiken et al., 2008). An update procedure to determine the elemental composition is reported by Canagaratna et al. (2015), showing the O/C and H/C derived from the method of Aiken et al. (2008) may be underestimated. The H/C and O/C were also derived using the newer approach by Canagaratna et al. (2015) and compared with the data derived from the Aiken et al. (2007) method. The H/C values derived using the Canagaratna et al. (2015) method strongly correlated with the values derived using Aiken et al. (2007) method (Fig. S1) and just increased by 27% as suggested by Canagaratna et al. (2015). Similar results were found for O/C and there was just a difference of 11% in O/C. Since only relative difference in elemental composition of SOA is studied here, only the data derived using Aiken et al. (2007) method are shown as the conclusion was not affected by the methods chosen. The fractional contribution of organics in the signals at $m/z$=44 and $m/z$=43 to total organics ($f_{44}$ and $f_{43}$, respectively) were also derived.

SOA yields were calculated as the ratio of organic aerosol mass formed to the amount of VOC reacted. The mass concentration of organic aerosol was derived using the total aerosol volume concentration measured by SMPS multiplied by the volume fraction of organics with a density of 1 g $cm^{-3}$ to better compare with previous literature. In the experiments with added $SO_2$, sulfuric acid was formed upon photooxidation and partly neutralized by background ammonia, which was introduced into the chamber mainly due to humidification. The volume fraction of organics was derived based on volume additivity using the mass of organics and ammonium sulfate/ammonium bisulfate from AMS and their respective density (1.32 g $cm^{-3}$ for organic aerosol from one of our previous studies (Flores et al., 2014) and the literature (Ng et al., 2007) and ~1.77 g $cm^{-3}$ for ammonium sulfate/ammonium bisulfate). According to the calculations based on the E-AIM model (Clegg et al., 1998;

Wexler and Clegg, 2002) (http://www.aim.env.uea.ac.uk/aim/ aim.php), there was no aqueous phase formed at
the relative humidity in the experiments of this study. The average RH for the period of monoterpene
photooxidation was 28-34% except for one experiment with average RH of 42% RH. The organic aerosol
concentration was corrected for the particle wall loss and dilution loss using the method described in Zhao et al.
(2015b).

## 153     2.2     Experimental procedure

The SOA formation from α-pinene and limonene photooxidation was investigated at different $NO_x$ and

$SO_2$ levels. Four types of experiments were done: with neither $NO_x$ nor $SO_2$ added (referred to as "low $NO_x$, low
$SO_2$"), with only $NO_x$ added (~ 20 ppb NO, referred to as "high $NO_x$, low $SO_2$"), with only $SO_2$ added (~15 ppb,
referred to as "low $NO_x$, high $SO_2$"), and with both $NO_x$ and $SO_2$ added (~20 ppb NO and ~15 ppb $SO_2$, referred
to as "high $NO_x$, high $SO_2$"). For low $NO_x$ conditions, background NO concentrations were around 0.05-0.2 ppb,
and NO was mainly from the background photolytic process of Teflon chamber wall (Rohrer et al., 2005). For
low $SO_2$ conditions, background $SO_2$ concentrations were below the detection limit of the $SO_2$ analyzer (0.05
ppb). In some experiments, a lower level of $SO_2$ (2 ppb, referred to as "moderate $SO_2$") was used to test the effect
of $SO_2$ concentration. An overview of the experiments is shown in Table 1.

In a typical experiment, the chamber was humidified to ~75% RH first, and then VOC and NO, if

applicable, were added to the chamber. Then the roof was opened to start photooxidation. In the experiments with
$SO_2$, $SO_2$ was added and the roof was opened to initialize nucleation first and then VOC was added. The particle
number concentration caused by $SO_2$ oxidation typically reached several $10^4$ $cm^{-3}$ (see Fig. 2 high $SO_2$ cases) and
after VOC addition, no further nucleation occurred. Adding $SO_2$ first and initializing nucleation by $SO_2$
photooxidation ensured that enough nucleating particles were present when VOC oxidation started. $SO_2$
concentration decayed slowly in the experiments with $SO_2$ added and most of the $SO_2$ was still left (typically
around 8 ppb from initial 15 ppb) at the end of an experiment due to its low reactivity with OH. Typical $SO_2$
time series in high $SO_2$ experiments are shown in Fig S2. The detailed conditions of the experiments are shown in
Table S1. The experiments of α-pinene and limonene photooxidation were designed to keep the initial OH
reactivity and thus OH loss rate constant so that the OH concentrations of these experiments were more
comparable. Therefore, the concentration of limonene was around one-third of the concentration of α-pinene due
to the higher OH reactivity of limonene.

## 176     2.3     Wall loss of organic vapors

The loss of organic vapors on chamber walls can influence SOA yield (Kroll et al., 2007; Zhang et al.,

2014; Ehn et al., 2014; Sarrafzadeh et al., 2016; McVay et al., 2016; Nah et al., 2016; Matsunaga and Ziemann,
2010; Ye et al., 2016; Loza et al., 2010). The wall loss rate of organic vapors in our chamber was estimated by
following the decay of organic vapor concentrations after photooxidation was stopped in the experiments with
low particle surface area (~$5\times10^{-8}$ $cm^2$ $cm^{-3}$) and thus low condensational sink on particles. Such method is
similar to the method used in previous studies (Ehn et al., 2014; Sarrafzadeh et al., 2016; Krechmer et al., 2016;
Zhang et al., 2015). A high-resolution time-of-flight chemical ionization mass spectrometer (HR-ToF-CIMS,
Aerodyne Research Inc.) with nitrate ion source ($^{15}NO_3^-$) was used to measure semi/low-volatile organic vapors.
The details of the instrument were described in our previous studies (Ehn et al., 2014; Sarrafzadeh et al., 2016).
The decay of vapors started from the time when the roof of the chamber was closed. The data were acquired at a
time resolution of 4 s. A typical decay of low-volatile organics is shown in Fig. S3 and the first-order wall loss
rate was determined to be around $6 \times 10^{-4}$ $s^{-1}$.
The SOA yield was not directly corrected for the vapor wall loss, but the influence of vapor wall loss on
SOA yield was estimated using the method in the study of Sarrafzadeh et al. (2016) and the details of the method
are described therein. Briefly, particle surface and chamber walls competed for the vapor loss (condensation) and
the condensation on particles led to particle growth. The fraction of organic vapor loss to particles in the sum of
the vapor loss to chamber walls and to particles ($F_p$) was calculated. The vapor loss to chamber walls was derived
using the wall loss rate. The vapor loss to particles was derived using particle surface area concentration,
molecular velocity and an accommodation coefficient $\alpha_p$ (Sarrafzadeh et al., 2016). $1/F_p$ ($f_{corr}$) provides the
correction factor to obtain the "real" SOA yield. $f_{corr}$ is a function of particle surface area concentration and
accommodation coefficient as shown in Fig. S4. Here a range of 0.1-1 for $\alpha_p$ was used, which is generally in line
with the ranges of $\alpha_p$ found by Nah et al. (2016) by fitting a vapor-particle dynamic model to experimental data.
At a given $\alpha_p$, the higher particle surface area, the lower $f_{corr}$ and the lower the influence of vapor wall loss are
because most vapors condense on particle surface and vice versa. At a given particle surface area, $f_{corr}$ decreases
with $\alpha_p$ because at higher $\alpha_p$ a larger fraction of vapors condenses on particles. An average molecular weight of
200 g/mol was used to estimate the influence of vapor wall loss. For the aerosol surface area range in most of the
experiments in this study (larger than $3 \times 10^{-6}$ $cm^2$ $cm^{-3}$), $f_{corr}$ is less than 1.4 (Fig. S4) and thus the influence of
vapor wall loss on SOA yield was relatively small ($< \sim 40\%$). Yet, for the experiments at high $NO_x$ and low $SO_2$
for $\alpha$-pinene and limonene, the influence of vapor wall loss on SOA can be high due to the low particle surface
area, especially at lower $\alpha_p$. We did not directly correct SOA yield for vapor wall loss because the correction
factor ($f_{corr}$) curve in the low surface area range is very steep and has very large uncertainties (Fig. S4). In
addition, $\alpha_p$ also has uncertainties and may depend on the identity of each condensable compounds.
**3  Results and discussion**
**3.1      Chemical scheme: VOC oxidation pathway and RO$_2$ fate**
In the photooxidation of VOC, OH and $O_3$ often co-exist and both contribute to VOC oxidation because $O_3$
formation in chamber studies is often unavoidable during photochemical reactions of VOC even in the presence
of trace amount of $NO_x$. In order to study the mechanism of SOA formation, it is helpful to isolate one oxidation
pathway from the other. In this study, the reaction rates of OH and ozone with VOC are quantified using
measured OH and $O_3$ concentrations multiplied by rate constants (time series of VOC, OH, and $O_3$ are shown in
Fig. S5). Typical OH and $O_3$ concentrations in an experiment were around $(1-15) \times 10^6$ molecules $cm^{-3}$ and 0-50
ppb, respectively, depending on the VOC and $NO_x$ concentrations added. For all the experiment in this study, the
VOC loss was dominated by OH oxidation over ozonolysis (see Fig. S6 as an example). The relative importance
of the reaction of OH and $O_3$ with monoterpenes was similar in the low $NO_x$ and high $NO_x$ experiments. At high
$NO_x$, OH was often higher while more $O_3$ was also produced. The dominant role of OH oxidation in VOC loss
makes the chemical scheme simple and it is easier to interpret than cases when both OH oxidation and ozonolysis
are important.

As mentioned above, $RO_2$ fate, i.e., the branching of $RO_2$ loss among different pathways, has an important

influence on the product distribution and thus on SOA composition, physicochemical properties, and yields. $RO_2$
can react with NO, $HO_2$, $RO_2$, or isomerize. The fate of $RO_2$ mainly depends on the concentrations of NO, $HO_2$
and $RO_2$. Here, the loss rates of $RO_2$ via different pathways were quantified using the measured $HO_2$, NO and
$RO_2$ concentrations and the rate constants based on the MCM3.3 (Jenkin et al., 1997; Saunders et al., 2003)
(*http://mcm.leeds.ac.uk/MCM.*). Measured $HO_2$ and $RO_2$ concentrations are shown in Fig. S7 as an example and
the relative importance of different $RO_2$ reaction pathways is compared in Fig. 1, which is similar for both α-
pinene and limonene oxidation. In the low $NO_x$ conditions of this study, $RO_2$+NO dominated the $RO_2$ loss rate in
the beginning of an experiment (Fig. 1a). The trace amount of NO (up to ~0.2 ppbV) was from the photolysis of
HONO, which was continuously produced from a photolytic process on chamber walls throughout an experiment
(Rohrer et al., 2005). But later in the experiment, $RO_2$+$HO_2$ contributed a significant fraction (up to ~40 %) to
$RO_2$ loss because of increasing $HO_2$ concentration and decreasing NO concentration. In the high $NO_x$ conditions,
$RO_2$+NO overwhelmingly dominated the $RO_2$ loss rate (Fig. 1b), and with the decrease of NO in an experiment,
the total $RO_2$ loss rate decreased substantially (Fig. 1b). Since the main products of $RO_2$+$HO_2$ are organic
hydroperoxides, more organic hydroperoxides relative to organic nitrates are expected in the low $NO_x$ conditions
here. The loss rate of $RO_2$+$RO_2$ was estimated to be ~$10^{-4}$ $s^{-1}$ using a reaction rate constant of $2.5 \times 10^{-13}$
molecules$^{-1}$ $cm^3$ $s^{-1}$ (Ziemann and Atkinson, 2012). This contribution is negligible compared to other pathways in
this study, although the reaction rate constants of $RO_2$+$RO_2$ are highly uncertain and may depend on specific $RO_2$
(Ziemann and Atkinson, 2012). Note that the $RO_2$ fate in the low and high $NO_x$ conditions quantified here are
further used in the discussion below since the information of $RO_2$ fate is important for data interpretation of
experiments conducted at different $NO_x$ levels (Wennberg, 2013).

**3.2    Effects of $NO_x$ and $SO_2$ on new particle formation**

The effects of $NO_x$ and $SO_2$ on new particle formation from α-pinene oxidation are shown in Fig. 2a. In

low $SO_2$ conditions, both the total particle number concentration and nucleation rate at high $NO_x$ were lower than
those at low $NO_x$, indicating $NO_x$ suppressed the new particle formation. The suppressing effect of $NO_x$ on new
particle formation was in agreement with the findings of Wildt et al. (2014). This suppression is considered to be
caused by the increased fraction of $RO_2$+NO reaction, decreasing the importance of $RO_2$+$RO_2$ permutation
reactions. $RO_2$+$RO_2$ reaction products are believed to be involved in the new particle formation (Wildt et al.,
2014; Kirkby et al., 2016) and initial growth of particles by forming higher molecular weight products such as
highly oxidized multifunctional molecules (HOM) and their dimers and trimers (Ehn et al., 2014; Kirkby et al.,
2016). Although the contribution of $RO_2$+$RO_2$ reaction to the total $RO_2$ loss is negligible, it can contribute a lot to
the compounds responsible for nucleation such as dimers and trimers. Generally, organic nitrates and primary
organic peroxides (from $RO_2(C_{10})$+$HO_2$) are not expected to be the main compounds responsible for nucleation
since although the volatility of these compounds is low (see below Sect. 3.3.1), it is likely not low enough to
nucleate.
In high $SO_2$ conditions, the nucleation rate and total number concentrations were high, regardless of $NO_x$
levels. The high concentration of particles was attributed to the new particle formation induced by $H_2SO_4$ alone
formed by $SO_2$ oxidation since the new particle formation occurred before VOC addition. The role of $H_2SO_4$ in
new particle formation has been well studied in previous studies (Berndt et al., 2005; Zhang et al., 2012; Sipila et
al., 2010; Kirkby et al., 2011; Almeida et al., 2013).
Similar suppression of new particle formation by $NO_x$ and enhancement of new particle formation by
$SO_2$ photooxidation were found for limonene oxidation (Fig. 2b).

### 3.3    Effects of $NO_x$ and $SO_2$ on SOA mass yield

#### 3.3.1    Effect of $NO_x$

Figure 3a shows SOA yield at different $NO_x$ for α-pinene oxidation. In order to make different
experiments more comparable, the SOA yield is plotted as a function of OH dose instead of reaction time. In low
$SO_2$ conditions, $NO_x$ not only suppressed the new particle formation but also suppressed SOA mass yield.
Because $NO_x$ suppressed new particle formation, the suppression of the SOA yield could be attributed to the lack
of new particles as seed and thus the lack of condensational sink, or to the decrease of condensable organic
materials. We further found that when new particle formation was already enhanced by added $SO_2$, the SOA yield
at high $NO_x$ was comparable to that at low $NO_x$ and the difference in SOA yield between high $NO_x$ and low $NO_x$
was much smaller (Fig. 3a). This finding can be attributed to two possible explanations. Firstly, $NO_x$ did not
significantly suppress the formation of low volatile condensable organic materials, although $NO_x$ obviously
suppressed the formation of products for nucleation. Secondly, $NO_x$ did suppress the formation of low-volatility
condensable organic materials via forming potentially more volatile compounds and in addition to that, the
suppressed formation of condensable organic materials was compensated by the presence of $SO_2$, resulting in
comparable SOA yield. Organic nitrates are a group of compounds formed at high $NO_x$, which have been
proposed to be more volatile (Presto et al., 2005; Kroll et al., 2006). However, many organic nitrates formed by
photooxidation in this study were highly oxidized organic molecules (HOMs) containing multi-functional groups
besides nitrate group ($C_{7-10}H_{9-15}NO_{8-15}$). These compounds are expected to have low volatility and they are found
to have an uptake coefficient on particles of ~1 (Pullinen et al., in preparation).  Therefore, the suppressing effect
of $NO_x$ on SOA yield was mostly likely due to suppressed nucleation, i.e., the lack of particle surface as
condensational sink. Due to the low particle surface area, the wall loss of condensable vapors in the experiment at
high $NO_x$ and low $SO_2$ was large (as shown by the large $f_{corr}$ in Fig. S4) and therefore SOA mass yield was
suppressed. If vapor wall loss is considered, the difference between the SOA yield at high $NO_x$ and at low $NO_x$
under low $SO_2$ conditions will be much reduced, as we found for high $SO_2$ cases (Fig. 3a). Under high $SO_2$
conditions, the influence of vapor wall loss on the difference in SOA yield between high $NO_x$ and low $NO_x$ was
minor (1%-8%, Fig. S4) due to the larger particle surface area.
For limonene oxidation, similar results of $NO_x$ suppressing the particle mass formation have been found
in low $SO_2$ conditions (Fig. 3b). Yet, in high $SO_2$ conditions, the SOA yield from limonene oxidation at high $NO_x$
was still significantly lower than that at low $NO_x$, which is different from the findings for α-pinene SOA. The
cause of this difference is currently unknown. Our data of SOA yield suggest that the products formed from
limonene oxidation at high $NO_x$ seemed to have higher average volatility than that at low $NO_x$.

The suppression of SOA mass formation by $NO_x$ under low $SO_2$ conditions agrees with previous studies
(Eddingsaas et al., 2012a; Wildt et al., 2014; Sarrafzadeh et al., 2016; Hatakeyama et al., 1991). For example, it
was found that high concentration of $NO_x$ (tens of ppb) suppressed mass yield of SOA formed from
photooxidation of β-pinene, α-pinene and VOC emitted by Mediterranean trees (Wildt et al., 2014; Sarrafzadeh et
al., 2016). And on the basis of the results by Eddingsaas et al. (2012a), the SOA yield at high $NO_x$ (referred to as
"high NO" by the authors) is lower than at low $NO_x$ in the absence of seed aerosol.

Our finding that the difference in SOA yield between high $NO_x$ and low $NO_x$ conditions was highly
reduced at high $SO_2$ is also in line with the findings of some previous studies using seed aerosols (Sarrafzadeh et
al., 2016; Eddingsaas et al., 2012a). For example, Sarrafzadeh et al. (2016) found that in the presence of seed
aerosol, the suppressing effect of $NO_x$ on the SOA yield from β-pinene photooxidation is substantially diminished
and SOA yield only decreases by 20-30% in the $NO_x$ range of <1 ppb to 86 ppb at constant OH concentrations.
The data by Eddingsaas et al. (2012a) also showed that in presence of seed aerosol, the difference in the SOA
yield between low $NO_x$ and high $NO_x$ is much decreased. However, our finding is in contrast with the findings in
other studies (Presto et al., 2005; Ng et al., 2007; Han et al., 2016; Stirnweis et al., 2017), who reported much
lower SOA yield at high $NO_x$ than at low $NO_x$ in presence of seed. The different findings in these studies from
ours may be attributed to the difference in the reaction conditions such as VOC oxidation pathways (OH
oxidation vs. ozonolysis), VOC and $NO_x$ concentration ranges, $NO/NO_2$, OH concentrations as well as organic
aerosol loading, which all affect SOA yield. The reaction conditions of this study often differ from those
described in the literature (see Table S2).

The difference in these conditions can result in both different apparent dependence on specific
parameters and the varied SOA yield. For example, SOA yield from α-pinene photooxidation at low $NO_x$ in this
study appeared to be much lower than that in Eddingsaas et al. (2012a). The difference between the SOA yield in
this study and some of previous studies and between the values in the literature can be attributed to several
reasons: 1) $RO_2$ fates may be different. For example, in our study at low $NO_x$, $RO_2$+NO account for a large
fraction of $RO_2$ loss while in Eddingsaas et al. (2012a) $RO_2$+$HO_2$ is the dominant pathway of $RO_2$ loss. This
difference in $RO_2$ fates may affect oxidation products distribution. 2) The organic aerosol loading of this study is
much lower than that some of previous studies (e.g., Eddingsaas et al. (2012a)) (see Fig. S9). SOA yields in this
study were also plotted versus organic aerosol loading to better compare with previous studies (Fig. S8 and S9).
3) The total particle surface area in this study may also differ from previous studies, which may influence the
apparent SOA yield due to vapor wall loss (the total particle surface area is often not reported in many previous
studies to compare with). 4) RH of this study is different from many previous studies, which often used very low
RH (<10%). It is important to emphasize that reaction conditions including the $NO_x$ as well as $SO_2$ concentration
range and RH in this study were chosen to be relevant to the anthropogenic-biogenic interactions in the ambient
atmosphere. In addition, difference in the organic aerosol density used in yield calculation should be taken into
account. In this study, SOA yield was derived using a density of 1 g $cm^{-3}$ to better compare with many previous
studies (e.g., (Henry et al., 2012)), while in some other studies SOA yield was derived using different density
(e.g., 1.32 g $cm^{-3}$ in Eddingsaas et al. (2012a)).

### 3.3.2 Effect of SO₂

For both α-pinene and limonene, SO₂ was found to enhance the SOA mass yield at given NO$_x$ levels, especially for the high NO$_x$ cases (Fig. 3). The enhancing effect of SO₂ on particle mass formation can be attributed to two reasons. Firstly, SO₂ oxidation induced new particle formation, which provided more surface and volume for further condensation of organic vapors. This is consistent with the finding that the enhancement of SOA yield by SO₂ was more significant at high NO$_x$ when the enhancement in nucleation was also more significant. Secondly, $H_2SO_4$ formed by photooxidation of SO₂ can enhance SOA formation via acid-catalyzed heterogeneous uptake, an important SOA formation pathway initially found from isoprene photooxidation (Jang et al., 2002; Lin et al., 2012; Surratt et al., 2007) and later also in the photooxidation of other compound such as anthropogenic VOC (Chu et al., 2016; Liu et al., 2016). For the products from monoterpene oxidation, such an acid-catalyzed effect may also occur (Northcross and Jang, 2007; Wang et al., 2012; Lal et al., 2012; Zhang et al., 2006; Ding et al., 2011; Iinuma et al., 2009) and in this study, the particles were acidic with the molar ratio of $NH_4^+$ to $SO_4^{2-}$ around 1.5-1.8, although no aqueous phase was formed.

We found that the SOA yield in the limonene oxidation at a moderate SO₂ level (2 ppb) was comparable to the yield at high SO₂ (15 ppb) when similar particle number concentrations in both cases were formed. Both yields were significantly higher than the yield at low SO₂ (<0.05 ppb, see Fig. S10). This comparison suggests that the effect in enhancing new particle formation by SO₂ seems to be more important compared to the particle acidity effect. The role of SO₂ in new particle formation is similar to adding seed aerosol and providing particle surface for organics to condense. Artificially added seed aerosol has been shown to enhance SOA formation from α-pinene and β-pinene oxidation (Ehn et al., 2014; Sarrafzadeh et al., 2016; Eddingsaas et al., 2012a). In some other studies, it was found that the SOA yield from α-pinene oxidation is independent of initial seed surface area (McVay et al., 2016; Nah et al., 2016). The difference in the literature may be due to the range of total surface area of particles, reaction conditions and chamber setup. For example, the peak particle-to-chamber surface ratio for α-pinene photooxidation in this study was $7.7\times10^{-5}$ at high NO$_x$ and low SO₂, much lower than the aerosol surface area range in the studies by Nah et al. (2016) and McVay et al. (2016). A lower particle-to-chamber surface ratio can lead to a larger fraction of organics lost on chamber walls. Hence, providing additional particle surface by adding seed particles can increase the condensation of organics on particles and thus increase SOA yield. However, once the surface area is high enough to inhibit condensation of vapors on chamber walls, further enhancement of particle surface will not significantly enhance the yield (Sarrafzadeh et al., 2016).

As mentioned above, the SOA yield at high NO$_x$ and low SO₂ was significantly suppressed due to vapor wall loss. If the influence of vapor wall loss is considered, the SOA yield at high NO$_x$ and low SO₂ will be much higher and thus the observed enhancement of SOA yield by SO₂ under high NO$_x$ conditions will be much less pronounced. Under low NO$_x$ conditions, the influence of vapor wall loss on the difference in SOA yield between high SO₂ and low SO₂ was minor (1%-7% for α-pinene and 5-32% for limonene, see Fig. S4) due to the larger particle surface area.

Particle acidity may also play a role in affecting the SOA yield in the experiments with high SO₂. Particle acidity was found to enhance the SOA yield from α-pinene photooxidation at high NO$_x$ (Offenberg et al., 2009) and "high NO" conditions (Eddingsaas et al., 2012a). Yet, in low NO$_x$ condition, particle acidity was reported to have no significant effect on the SOA yield from α-pinene photooxidation (Eddingsaas et al., 2012a; Han et al.,

2016). According to these findings, at low $NO_x$ the enhancement of SOA yield in this study is attributed to the
effect of facilitating nucleation and providing more particle surface by $SO_2$ photooxidation. At high $NO_x$, the
effect in enhancing new particle formation by $SO_2$ photooxidation seems to be more important, although the
effect of particle acidity resulted from $SO_2$ photooxidation may also play a role.

$SO_2$ has been proposed to also affect gas phase chemistry of organics by changing the $HO_2/OH$ or

forming $SO_3$ (Friedman et al., 2016). In this study, the effect of $SO_2$ on gas phase chemistry of organics was not
significant because of the much lower reactivity of $SO_2$ with OH compared with α-pinene and limonene
(Atkinson et al., 2004, 2006; Atkinson and Arey, 2003) and the low OH concentrations (2-3 orders of magnitude
lower than those in the study by Friedman et al. (2016)). Moreover, reactions of $RO_2$ with $SO_2$ was also not
important because the reaction rate constant is very low ($<10^{-14}$ molecule$^{-1}$ cm$^3$ s$^{-1}$) (Lightfoot et al., 1992; Berndt
et al., 2015). In addition, from the AMS data of SOA formed at high $SO_2$ no significant organic fragments
containing sulfur were found. Also the fragment $CH_3SO_2^+$ from organic sulfate suggested by Farmer et al. (2010)
was not detected in our data. The absence of organic sulfate tracers is likely due to the lack of aqueous phase in
aerosol particles in this study. Therefore, the influence of $SO_2$ on gas phase chemistry of organics and further on
SOA yield via affecting gas phase chemistry is not important in this study.

The presence of high $SO_2$ enhanced the SOA mass yield at high $NO_x$ conditions, which was even

comparable with the SOA yield at low $NO_x$ for α-pinene oxidation. This finding indicates that the suppressing
effect of $NO_x$ on SOA mass formation was compensated to large extent by the presence of $SO_2$. This has
important implications for SOA formation affected by anthropogenic-biogenic interactions in the real atmosphere
when $SO_2$ and $NO_x$ often co-exist in relative high concentrations as discussed below.

### 3.4    Effects of $NO_x$ and $SO_2$ on SOA chemical composition

The effects of $NO_x$ and $SO_2$ on SOA chemical composition were analyzed on the basis of AMS data. We

found that $NO_x$ enhanced nitrate formation. The ratio of the mass of nitrate to organics was higher at high $NO_x$
than at low $NO_x$ regardless of the $SO_2$ level, and similar trends were found for SOA from α-pinene and limonene
oxidation (Fig. 4a). Higher nitrate to organics ratios were observed for SOA from limonene at high $NO_x$, which is
mainly due to the lower $VOC/NO_x$ ratio resulted from the lower concentrations of limonene (7 ppb) compared to
α-pinene (20 ppb) (see Table 1). Overall, the mass ratios of nitrate to organics ranged from 0.02 to 0.11
considering all the experiments in this study.

Nitrate formed can be either inorganic (such as $HNO_3$ from the reaction of $NO_2$ with OH) or organic (from

the reaction of $RO_2$ with NO). The ratio of $NO_2^+$ ($m/z$=46) to $NO^+$ ($m/z$=30) in the mass spectra detected by AMS
can be used to differentiate whether nitrate is organic or inorganic (Fry et al., 2009; Rollins et al., 2009; Farmer et
al., 2010; Kiendler-Scharr et al., 2016). Organic nitrate was considered to have a $NO_2^+/NO^+$ of ~0.1 and inorganic
$NH_4NO_3$ had a $NO_2^+/NO^+$ of ~0.31 with the instrument used in this study as determined from calibration
measurements. In this study, $NO_2^+/NO^+$ ratios ranged from 0.14 to 0.18, closer to the ratio of organic nitrate. The
organic nitrate was estimated to account for 57%-77% (molar fraction) of total nitrate considering both the low
$NO_x$ and high $NO_x$ conditions. This indicates that nitrate was mostly organic nitrate, even at low $NO_x$ in this
study.
In order to determine the contribution of organic nitrate to total organics, we estimated the molecular
weight of organic nitrates formed by α-pinene and limonene oxidation to be 200-300 g/mol, based on reaction
mechanisms ((Eddingsaas et al., 2012b) and MCM v3.3, via website: http://mcm.leeds.ac.uk/MCM.). We
assumed a molecular weight of 200 g/mol in order to make our results comparable to the field studies which used
similar molecular weight (Kiendler-Scharr et al., 2016). For this value, the organic nitrate compounds were
estimated to account for 7-26% of the total organics mass as measured by AMS in SOA. Organic nitrate fraction
in total organics was within the range of values found in a field observation in southeast US (5-12% in summer
and 9-25% in winter depending on the molecular weight of organic nitrate) using AMS (Xu et al., 2015b) and
particle organic nitrate content derived from the sum of speciated organic nitrates (around 1-17% considering
observed variability and 3% and 8% on average in the afternoon and at night, respectively) (Lee et al., 2016).
Note that the organic nitrate fraction observed in this study was lower than the mean value (42%) for a number of
European observation stations when organic nitrate is mainly formed by the reaction of VOC with $NO_3$
(Kiendler-Scharr et al., 2016).
Moreover, we found that the contribution of organic nitrate to total organics (calculated using a
molecular weight of 200 g/mol for organic nitrate) was higher at high $NO_x$ (Fig. 4b), although in some
experiments the ratios of $NO_2^+$ to $NO^+$ were too noisy to derive a reliable fraction of organic nitrate. This result is
consistent with the reaction scheme that at high $NO_x$, almost all $RO_2$ loss was switched to the reaction with NO,
which is expected to enhance the organic nitrate formation. Besides organic nitrate, the ratio of nitrogen to carbon
atoms (N/C) was also found to be higher at high $NO_x$ (Fig. S11). But after considering nitrate functional group
separately, N/C ratio was very low, generally <0.01, which indicates majority of the organic nitrogen existed in
the form of organic nitrate.
The chemical composition of organic components of SOA in terms of H/C and O/C ratios at different
$NO_x$ and $SO_2$ levels was further compared. For SOA from α-pinene photooxidation, in low $SO_2$ conditions, no
significant difference in H/C and O/C was found between SOA formed at low $NO_x$ and at high $NO_x$ within the
experimental uncertainties (Fig. 5). The variability of H/C and O/C at high $NO_x$ is large, mainly due to the low
particle mass and small particle size. In high $SO_2$ conditions, SOA formed at high $NO_x$ had the higher O/C and
lower H/C, which indicates that SOA components had higher oxidation state. The higher O/C at high $NO_x$ than at
low $NO_x$ is partly due to the higher OH dose at high $NO_x$, although even at same OH dose O/C at high $NO_x$ was
still slightly higher than at low $NO_x$ in high $SO_2$ conditions.
For the SOA formed from limonene photooxidation, no significant difference in the H/C and O/C was
found between different $NO_x$ and $SO_2$ conditions (Fig. S12), which is partly due to the low signal resulting from
low particle mass and small particle size in high $NO_x$ conditions.
Due to the high uncertainties for some of the H/C and O/C data, the chemical composition was further
analyzed using $f_{44}$ and $f_{43}$ since $f_{44}$ and $f_{43}$ are less noisy (Fig. 6). For both α-pinene and limonene, SOA formed at
high $NO_x$ generally had lower $f_{43}$. Because $f_{43}$ generally correlates with H/C in organic aerosol (Ng et al., 2011),
lower $f_{43}$ is indicative of lower H/C, which is consistent with the lower H/C at high $NO_x$ observed for SOA from
α-pinene oxidation in high $SO_2$ conditions (Fig. 5). The lower $f_{43}$ at high $NO_x$ was evidenced in the oxidation of
α-pinene based on the data in a previous study (Chhabra et al., 2011). The lower H/C and $f_{43}$ are likely to be
related to the reaction pathways. According to the reaction mechanism mentioned above, at low $NO_x$ a significant
fraction of $RO_2$ reacted with $HO_2$ forming hydroperoxides, while at high $NO_x$ almost all $RO_2$ reacted with NO
forming organic nitrates. Compared with organic nitrates, hydroperoxides have higher H/C ratio. The same
mechanism also caused higher organic nitrate fraction at high $NO_x$, as discussed above.
Detailed mass spectra of SOA were compared, shown in Fig 7. For α-pinene, in high $SO_2$ conditions,
mass spectra of SOA formed at high $NO_x$ generally had higher intensity for CHOgt1 ("gt1" means greater than 1)
family ions, such as $CO_2^+$ (*m/z* 44), but lower intensity for CH family ions, such as $C_2H_3^+$ (*m/z* 15), $C_3H_3^+$ (*m/z* 39)
(Fig. 7b) than at low $NO_x$. In low $SO_2$ conditions, such difference is not apparent (Fig. 7a), partly due to the low
signal from AMS for SOA formed at high $NO_x$ as discussed above. For both the high $SO_2$ and low $SO_2$ cases,
mass spectra of SOA at high $NO_x$ show higher intensity of CHN1 family ions. This is also consistent with the
higher N/C ratio shown above. For SOA from limonene oxidation, SOA formed at high $NO_x$ had lower mass
fraction at *m/z* 15 ($C_2H_3^+$), 28 ($CO^+$), 43 ($C_2H_3O^+$), 44 ($CO_2^+$), and higher mass fraction at *m/z* 27 ($CHN^+$, $C_2H_3^+$),
41 ($C_3H_5^+$), 55 ($C_4H_7^+$), 64 ($C_4O^+$) than at low $NO_x$ (Fig. S13). It seems that overall mass spectra of the SOA from
limonene formed at high $NO_x$ had higher intensity for CH family ions, but lower intensity for CHO1 family ions
than at low $NO_x$. Note that the differences in these *m/z* were based on the average spectra during the whole
reaction period and may not reflect the chemical composition at a certain time.

## 4 Conclusion and implications

We investigated the SOA formation from the photooxidation of α-pinene and limonene under different $NO_x$
and $SO_2$ conditions, when OH oxidation was the dominant oxidation pathway of monoterpenes. The fate of $RO_2$
was regulated by varying $NO_x$ concentrations. We confirmed that $NO_x$ suppressed new particle formation. $NO_x$
also suppressed SOA mass yield in the absence of $SO_2$. The suppression of SOA yield by $NO_x$ was likely due to
the suppressed new particle formation, i.e., absence of sufficient particle surfaces for organic vapor to condense
on at high $NO_x$, which could result in large vapor loss to chamber walls.
$SO_2$ enhanced SOA yield from α-pinene and limonene photooxidation. $SO_2$ oxidation produced high number
concentration of particles and compensated for the suppression of SOA yield by $NO_x$ to a large extent. The
enhancement of SOA yield by $SO_2$ is likely to be mainly caused by facilitating nucleation by $H_2SO_4$, although the
contribution of acid-catalyzed heterogeneous uptake cannot be excluded.
$NO_x$ promoted nitrate formation. The majority (57-77%) of nitrate was organic nitrate at both low $NO_x$ and
high $NO_x$, based on the estimate using the $NO_2^+/NO^+$ ratios from AMS data. The significant contribution of
organic nitrate to nitrate may have important implications for deriving the hygroscopicity from chemical
composition. For example, a number of studies derived the hygroscopicity parameter by linear combination of the
hygroscopicity parameters of various components such as sulfate, nitrate, and organics, assuming all nitrates are
inorganic nitrate (Wu et al., 2013; Cubison et al., 2008; Yeung et al., 2014; Bhattu and Tripathi, 2015; Jaatinen et
al., 2014; Moore et al., 2012; Gysel et al., 2007). Because the hygroscopicity parameter of organic nitrate may be
much lower than inorganic nitrate (Suda et al., 2014), such derivation may overestimate hygroscopicity.
Organic nitrate compounds are estimated to contribute 7-26% of the total organics using an average
molecular weight of 200 g/mol for organic nitrate compounds and a higher contribution of organic nitrate was
found at high $NO_x$. Generally, SOA formed at high $NO_x$ has a lower H/C compared to that at low $NO_x$. The
higher contribution of organic nitrate to total organics and lower H/C at high $NO_x$ than at low $NO_x$ is attributed to

the reaction of $RO_2$ with NO, which produced more organic nitrates relative to organic hydroperoxides formed via the reaction of $RO_2$ with $HO_2$. The different chemical composition of SOA between high and low $NO_x$ conditions may affect the physicochemical properties of SOA such as volatility, hygroscopicity, and optical properties and thus change the impact of SOA on environment and climate.

In this study, the influence of vapor wall loss on SOA yield was estimated although the SOA yields in this study were not corrected for vapor wall loss. We need to be cautious about the enhancement of the SOA yield by $SO_2$ under high $NO_x$ conditions and the suppression of the SOA yield by $NO_x$ under low $SO_2$ conditions. These effects will be less pronounced when vapor wall loss is considered because of the significant vapor loss to chamber walls rather than to particles at low particle surface concentration. Yet, the low particle surface concentration and thus low condensational sink of vapors to particle surface reflect some real cases in the atmosphere, because when the condensational sink by particle surface is low in the atmosphere, organic vapors will be lost to the next largest sink, e.g. dry deposition. Nevertheless, our important findings hold for the influence of $NO_x$ and $SO_2$ on SOA new particle formation, mass yield, and chemical composition, showing indeed the interaction of anthropogenic and biogenic emissions in the process of SOA formation.

The different effects of $NO_x$ and $SO_2$ on new particle formation and SOA mass yields have important implications for SOA formation affected by anthropogenic-biogenic interactions in the ambient atmosphere. When an air mass of anthropogenic origin is transported to an area enriched in biogenic VOC emissions or vice versa, anthropogenic-biogenic interactions occur. Such scenarios are common in the ambient atmosphere in many areas. For example, Kiendler-Scharr et al. (2016) shows that the organic nitrate concentrations are high in all the rural sites all over Europe, indicating the important influence of anthropogenic emissions in rural areas which are often enriched in biogenic emissions. [14]C analysis in several studies show that modern source carbon, from biogenic emission or biomass burning, account for large fractions of organic aerosol even in urban areas (Szidat et al., 2009; Weber et al., 2007; Sun et al., 2012), indicating the potential interactions of biogenic emissions with anthropogenic emissions in urban areas. In such cases, anthropogenic $NO_x$ alone may suppress the new particle formation and SOA mass from biogenic VOC oxidation, as we found in this study, because in principle the suppression of SOA mass due to suppressed nucleation can occur in the ambient atmosphere, although chamber experiments often cannot accurately simulate the vapor loss on surface in the boundary layer. However, due to the co-existence of $NO_x$ with $SO_2$, $H_2SO_4$ formed by $SO_2$ oxidation can counteract such suppression of particle mass because regardless of $NO_x$ levels, $H_2SO_4$ can induce new particle formation especially in the presence of water, ammonia or amine (Berndt et al., 2005; Zhang et al., 2012; Sipila et al., 2010; Almeida et al., 2013; Kirkby et al., 2011; Chen et al., 2012). The overall effects on SOA mass depend on specific $NO_x$, $SO_2$ and VOC concentrations and VOC types as well as anthropogenic aerosol concentrations and can be a net suppressing, neutral, or enhancing effect. Such scheme is depicted in Fig. 8. Other anthropogenic emissions, such as primary anthropogenic aerosol and precursors of anthropogenic secondary aerosol, can have similar roles as $SO_2$. By affecting the concentrations of $SO_2$, $NO_x$, and anthropogenic aerosol, anthropogenic emissions may have important mediating impacts on biogenic SOA formation. Considering the effects of these factors in isolation may cause bias in predicting biogenic SOA concentrations. The combined impacts of $SO_2$, $NO_x$, and anthropogenic aerosol are also important to the estimate on how much organic aerosol concentrations will change with the ongoing and future reduction of anthropogenic emissions (Carlton et al., 2010).

**Acknowledgements**

We thank the SAPHIR team, especially Rolf Häseler, Florian Rubach, Dieter Klemp for supporting our measurements and providing helpful data. M. J. Wang would like to thank China Scholarship Council for funding the joint PhD program.

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

Table 1 Overview of the experiments in this study

| Precursor | $SO_2$ | $NO_x$ | NO (ppb) | $SO_2$ (ppb) |
|---|---|---|---|---|
| α-pinene | Low $SO_2$ | Low $NO_x$ | 0.05-0.2 | <0.05 |
| | | High $NO_x$ | ~20 | <0.05 |
| (~20 ppb) | High $SO_2$ | Low $NO_x$ | 0.05-0.2 | ~15 |
| | | High $NO_x$ | ~20 | ~15 |
| Limonene | Low $SO_2$ | Low $NO_x$ | 0.05-0.2 | <0.05 |
| | | High $NO_x$ | ~20 | <0.05 |
| (~7 ppb) | High $SO_2$ | Low $NO_x$ | 0.05-0.2 | ~15 |
| | | High $NO_x$ | ~20 | ~15 |
| | Moderate $SO_2$ | High $NO_x$ | ~20 | ~2 |

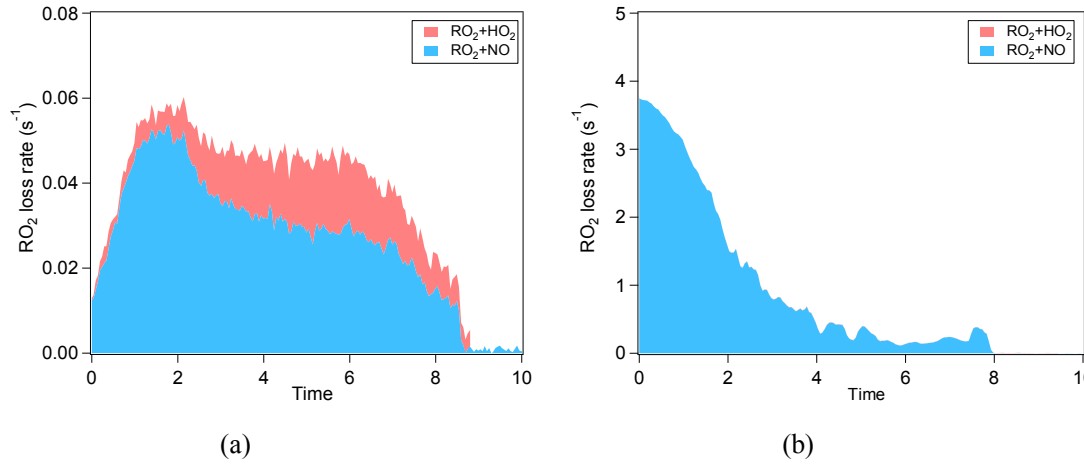


(a)                             (b)

Figure 1. Typical loss rate of $RO_2$ by $RO_2$+NO and $RO_2$+$HO_2$ in the low $NO_x$ (a) and the high $NO_x$ (b) conditions
of this study. The experiments at low $SO_2$ are shown. The $RO_2$+$HO_2$ rate is stacked on the $RO_2$+NO rate. Note
the different scales for $RO_2$ loss rate in panel a and b. In panel b, the contribution of $RO_2$+$HO_2$ is very low and
barely noticeable.

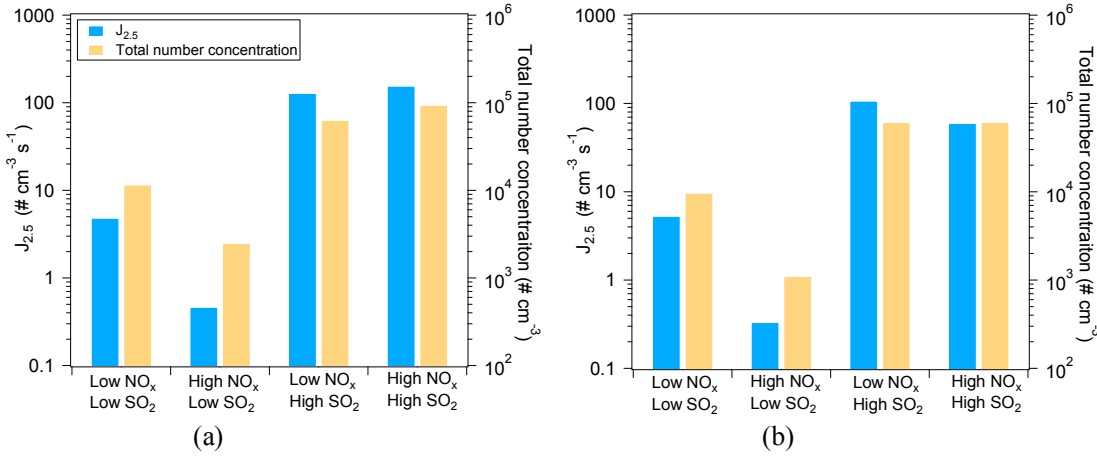

Figure 2. Nucleation rates ($J_{2.5}$) and maximum total particle number concentrations under different $NO_x$ and $SO_2$ conditions for the SOA from α-pinene oxidation (a) and from limonene oxidation (b).



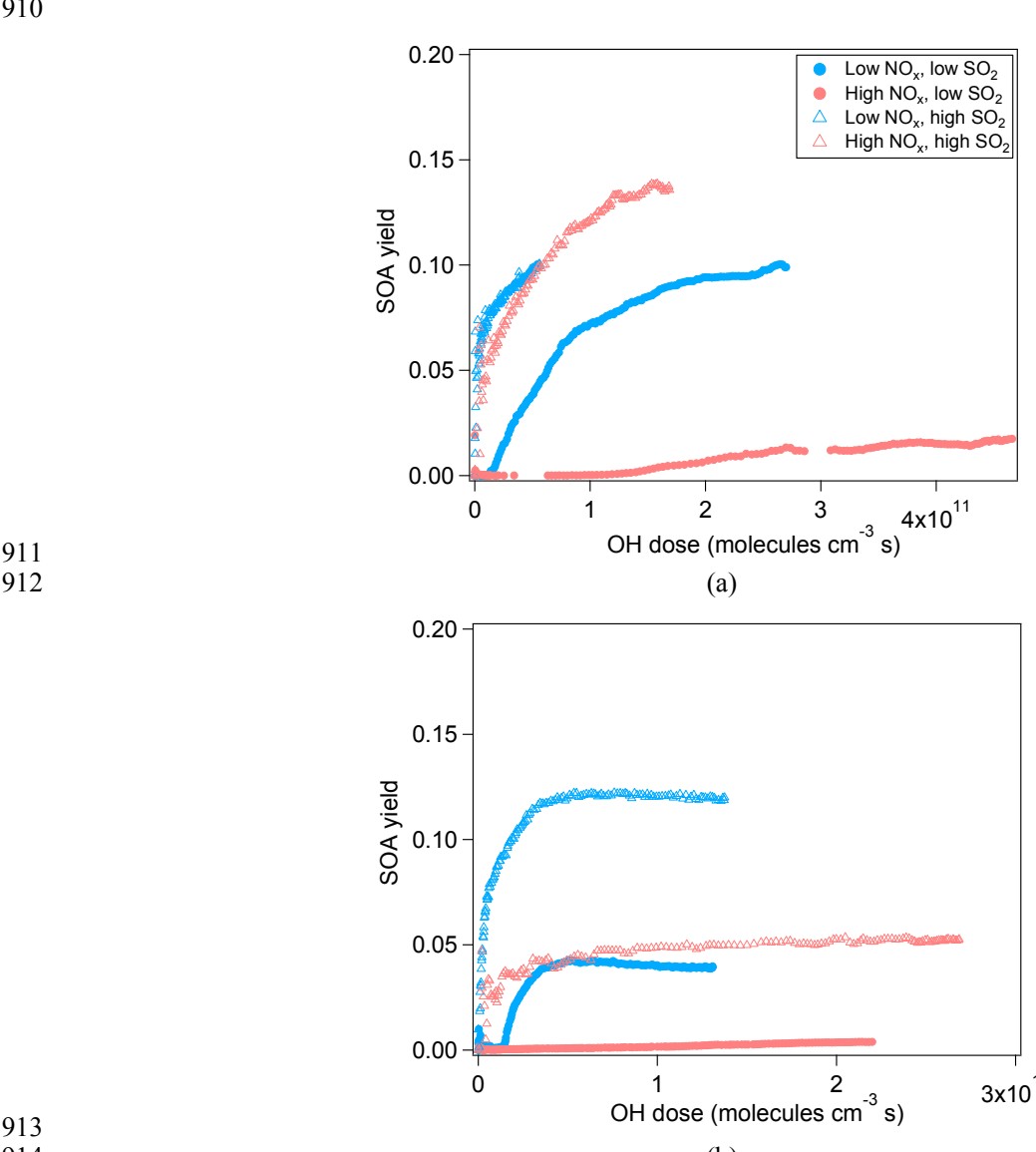


(a)


(b)
Figure 3. SOA yield of the photooxidation of α-pinene (a) and limonene (b) in different $NO_x$ and $SO_2$ conditions.

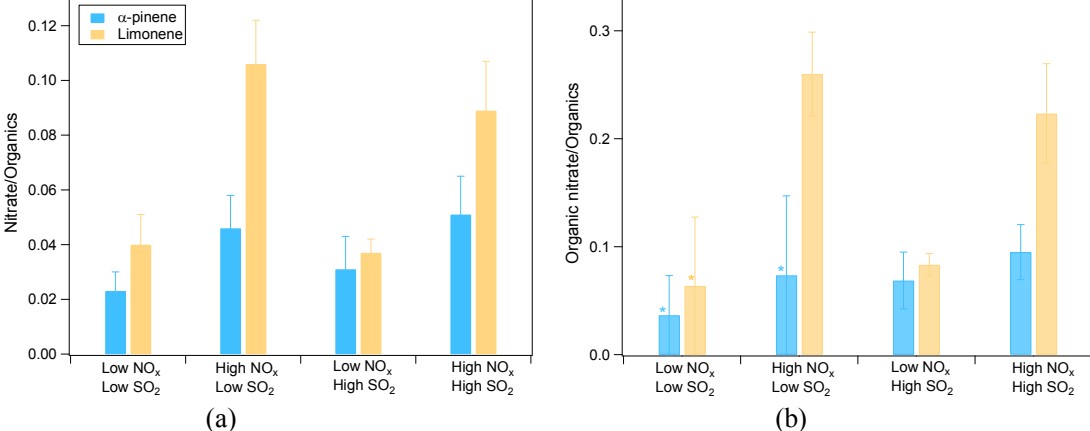

(a)                                    (b)

Figure 4. (a) The ratio of nitrate mass concentration to organics mass in different $NO_x$ and $SO_2$ conditions. The
average ratios of nitrate to organics during the reaction are shown and error bars indicate the standard deviations.
(b) The fraction of organic nitrate to total organics in different $NO_x$ and $SO_2$ conditions calculated using a
molecular weight of 200 g/mol for organic nitrate. The average fractions during the reaction are shown and error
bars indicate the standard deviations. In panel b, * indicate the experiments where the ratios of $NO_2^+$ to $NO^+$ were
too noisy to derive a reliable fraction of organic nitrate. For these experiments, 50% of total nitrate was assumed
to be organic nitrate and the error bars show the range when 0 to 100% of nitrate are assumed to be organic
nitrate.

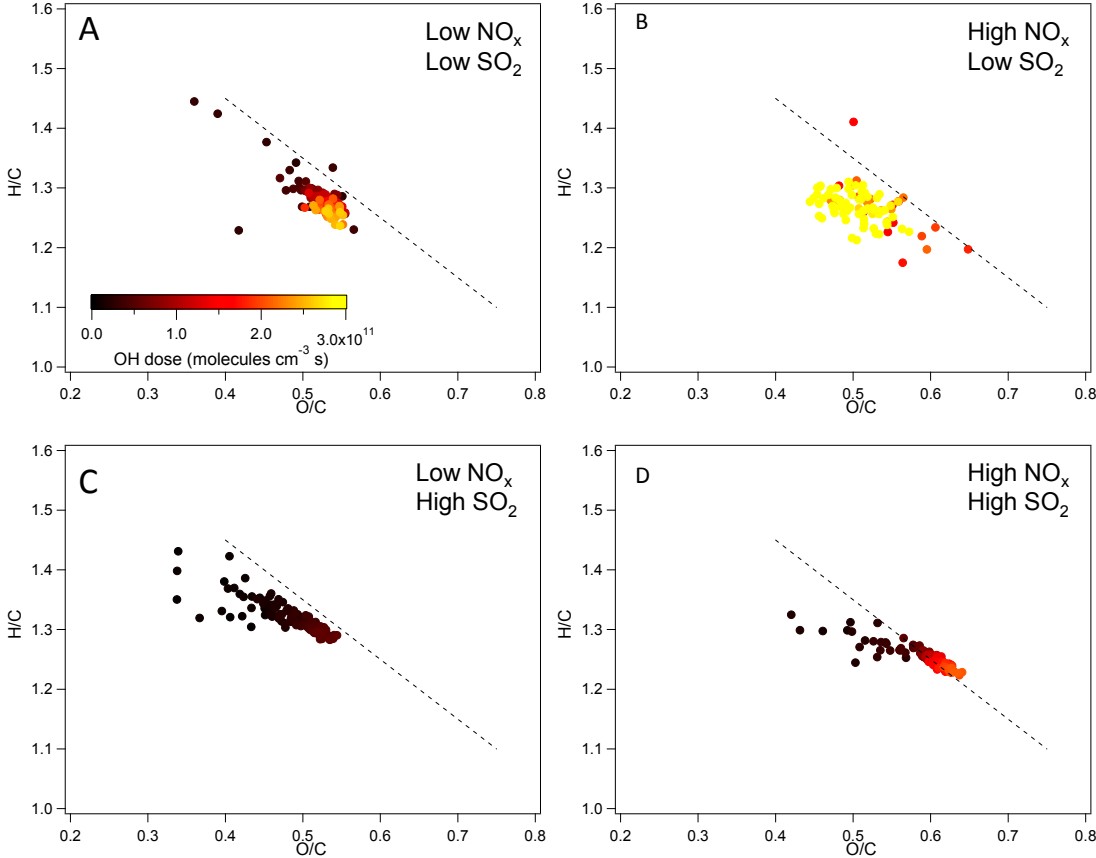

Figure 5. H/C and O/C ratio of SOA from photooxidation of α-pinene in different $NO_x$ and $SO_2$ conditions. A:
low $NO_x$, low $SO_2$, B: high $NO_x$, low $SO_2$, C: low $NO_x$, high $SO_2$, D: high $NO_x$, high $SO_2$. The black dashed line
corresponds to the slope of -1.

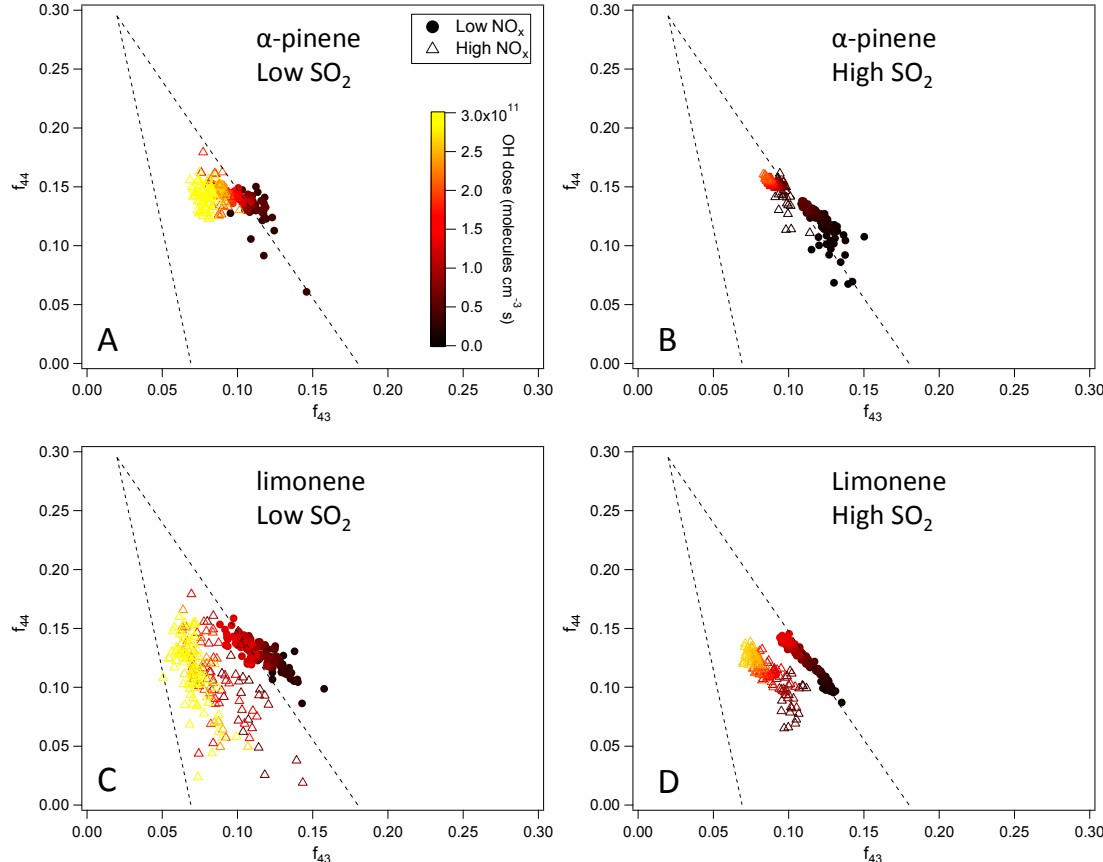

Figure 6. $f_{44}$ and $f_{43}$ of SOA from the photooxidation of α-pinene and limonene in different $NO_x$ and $SO_2$
conditions. A: α-pinene, low $SO_2$, B: α-pinene, high $SO_2$, C: limonene, low $SO_2$, D: limonene, high $SO_2$. Note
that in the low $SO_2$, high $NO_x$ condition (panel C), the AMS signal of SOA from limonene oxidation was too low
to derive reliable information due to the low particle mass concentration and small particle size. Therefore, the
data for high $NO_x$ in panel C show an experiment with moderate $SO_2$ (2 ppb) and high $NO_x$ instead.

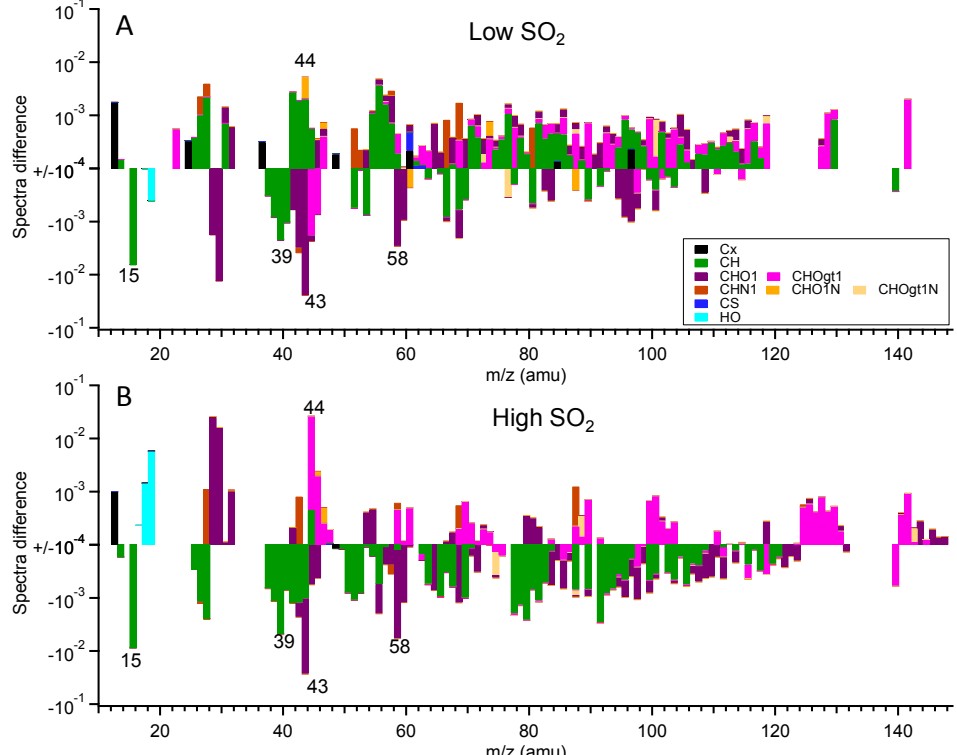


Figure 7. The difference in the mass spectra of organics of SOA from α-pinene photooxidation between high $NO_x$
and low $NO_x$ conditions (high $NO_x$-low $NO_x$). SOA was formed at low $SO_2$ (a) and high $SO_2$ (b). The different
chemical family of high resolution mass peaks are stacked at each unit mass *m/z* ("gt1" means greater than 1).
The mass spectra were normalized to the total organic signals. Note the log scale of y-axis and only the data with
absolute values large than $10^{-4}$ are shown.

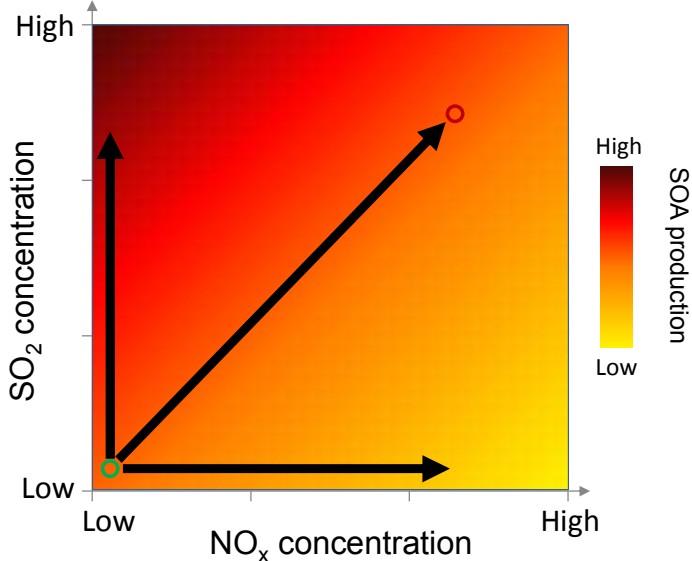


Figure 8. Conceptual schematic showing how $NO_x$ and $SO_2$ concentrations affect biogenic SOA mass production.
The darker colors indicate higher SOA production. The circle on the bottom left corner indicates biogenic cases
and the circle on the right top corner indicates the anthropogenic cases. And the horizontal and vertical arrows
indicate the effect of $NO_x$ and $SO_2$ alone. The overall effects on SOA production depend on specific $NO_x$, $SO_2$
concentrations and VOC concentrations and speciation.