# Peer review of "Effects of NOx and SO2 on the Secondary Organic Aerosol Formation from Photooxidation of α-pinene and Limonene"

_Atmospheric Chemistry and Physics, 2017_

## Referee Comment (RC1) · Anonymous Referee #1 · 26 Apr 2017

This manuscript describes the competing roles of NOx and SO2 on SOA formation of a-pinene and limonene. The ability of SO2 to enhance seed aerosol surface area appears to be a dominant factor, and that enhancing seed aerosol reduces the NOx suppression of SOA yields, at least in some monoterpenes. The authors use their AMS data to determine the role of organic nitrates in SOA, and find that organic nitrates account for a substantial fraction of the SOA mass. Overall, this is an interesting piece of work, and warrants publication in the ACP following some revision.

Major Comments

The nature of the experimental design was not so much to look at the impact of SO2 – but to look at the role of a sulfate seed aerosol. From the manuscript, my interpretation

is that the SO2 additions were used to nucleate (inorganic) seed aerosol. Was there any SO2 left over to impact VOC oxidation? It is not clear to me if the SO2 additions really paralleled the NOx additions, because the experimental design was different. That's not to say that these aren't valuable experiments that add to the literature! I merely question whether this was truly an 'SO2 addition' rather than a 'sulfate aerosol addition' to VOC oxidation experiments.

SOA yield is influenced strongly by OA mass. The authors plot SOA yield versus OH dose, which is certainly a useful figure to see – but it is hard to compare the SOA yields if the SOA mass has not bee accounted for. The authors need to also show SOA yield versus OA mass so that the readers can contrast the relationships to other studies. It would be useful to compare the SOA yields to other studies: how do the yield values compare to other measurements of OH oxidation of a-pinene? This will allow readers to place the studies in context.

The results of SO2 and NOx effects on SOA yield are consistent with the Sarrafzadeh and Eddingsaas studies, which found that the presence of seed aerosol suppresses the 'NOx effect' on SOA yield. However, they contradict previous studies (e.g. Ng et al. 2007, Presto et al. 2005). The authors need to do a better job of contrasting their studies – they attribute the difference to a vague collection of parameters (e.g. NO:NO2 ratio, OH concentrations, etc.). It would be extremely helpful if the authors could synthesize the information (i.e. put numbers on those parameters) to help readers understand the differences in experimental conditions across the studies. A table would be particularly helpful.

Lines 118: the use of the HR-ToF-AMS to derive elemental ratios uses the older Aiken method. However, as the authors note, the newer 2015 approach corrects some underestimation. Because readers may wish to compare results across studies in the future, it is appropriate and prudent to update the results to the newer calculations.

Line 128: the authors note that they account for particle wall loss and dilution loss, but

not for vapor wall loss. Recent papers have shown this to be a chemically-dependent and substantial effect on SOA yields, and most rigorous SOA yield work now accounts for these effects. How will ignoring vapor wall loss influence the results – and the interpretation thereof?

In the Introduction, the authors do a good job of summarizing the reasons why such a study would be interesting. Much of the discussion focuses on the role of NOx on SOA yields – this is reasonable as most of the literature has focused on that problem! However, there is some relatively recent literature regarding the role of SO2 in affecting SOA chemistry and monoterpene OH oxidation that the authors should consider. In particular:

Photooxidation of cyclohexene in the presence of SO2: SOA yield and chemical composition. Shijie Liu, Long Jia, Yongfu Xu, Narcisse T. Tsona, Shuangshuang Ge, and Lin Du. Atmos. Chem. Phys. Discuss., doi:10.5194/acp-2017-30, 2017

Synergetic formation of secondary inorganic and organic aerosol: effect of SO2 and NH3 on particle formation and growth. Biwu Chu, Xiao Zhang, Yongchun Liu, Hong He, Yele Sun, Jingkun Jiang, Junhua Li, and Jiming Hao. Atmos. Chem. Phys., 16, 14219-14230, doi:10.5194/acp-16-14219-2016, 2016

Formation of secondary aerosols from gasoline vehicle exhaust when mixing with SO2. T. Liu, X. Wang, Q. Hu, W. Deng, Y. Zhang, X. Ding, X. Fu, F. Bernard, Z. Zhang, S. Lü, Q. He, X. Bi, J. Chen, Y. Sun, J. Yu, P. Peng, G. Sheng, and J. Fu. Atmos. Chem. Phys., 16, 675-689, doi:10.5194/acp-16-675-2016, 2016

Anthropogenic Sulfur Perturbations on Biogenic Oxidation: SO2 Additions Impact Gas-Phase OH Oxidation Products of $\alpha$- and $\beta$-Pinene. Beth Friedman, Patrick Brophy, William H. Brune, and Delphine K. Farmer. Environmental Science & Technology 2016 50 (3), 1269-1279. DOI: 10.1021/acs.est.5b05010

Is there any evidence for organic sulfates in the SOA from the AMS data? This has

been a subject of some debate in the literature, and an additional datapoint would be useful. This may also clarify the role of acid catalysis, as I believe that has been linked to the formation of organic sulfates.

Minor Comments

Line 136. The authors note an average RH of 28-42% for the experiments. This seems like a relatively large range: will this affect the SOA yields, or interpretation of the data?

Re: Discussion of SO2 effects. The authors dominantly attribute the enhancement of SOA by SO2 to increased particle surface area, or perhaps to acid catalysis. These seem like extremely likely reasons; however, there is one study that suggests that SO2 will influence gas-phase oxidation products (Friedman et al.), which could also be a confounding factor unless all of the SO2 is in the particle phase before VOC oxidation commences... This would be a useful clarification.

Technical comments.

Line 26: should read "compared to low NOx"

Line 29: should read "SO2 can compensate for such effects"

Introduction: line 34: sentence has repetitive 'importants': consider removing at least one (e.g. "SOA is an important class of atmospheric aerosol" seems like an unnecessary statement for the journal's audience). This adjective is used heavily throughout the introduction (lines 45, 49), and I recommend removing or replacing the adjective to improve readability

Line 56: hydroperoxides should be plural

Line 57: need comma between 'NO' and 'forming'

Line 87: should read "might have either counteracting or synergistic effects on SOA..."

Line 126: remove the with following 'multiplied by'

Line 135: should read 'there was no aqueous..'

Line 221, remove comma between 'that' and 'high'

Line 360: should read 'in the ambient atmosphere'

---

## Referee Comment (RC2) · Anonymous Referee #2 · 18 Jun 2017

This chamber study investigated the effects of SO2 and NOx (NO) on SOA formation from photooxidation of a-pinene and limonene. It was found that SO2 enhanced SOA yield while NOx suppressed SOA yield. The suppression effect of NOx was attributed to the suppressed new particle formation and thus a lack of particle surface area for organics to condense on. The authors concluded that SO2 oxidation produced high number of particles and compensated for the suppression of SOA yield by NOx. SOA composition measured by AMS was also presented and discussed.

This is an interesting study. The gas- and particle-phase measurements are comprehensive and include several important species that have not been typically characterized in previous studies (e.g., OH, HO2 and RO2). The experiments appeared to be carefully conducted. However, I have major concerns regarding data interpretation and some conclusions in the manuscript.

One of the central themes of the manuscript is that the suppression effect of NOx on SOA formation can be compensated by the presence of SO2. This conclusion is not accurate based on all the data presented in this manuscript. For a-pinene, it appears that under high SO2 conditions, the SOA yields under low vs. high NOx conditions are comparable. However, this is not the case for limonene, where there is still a large difference in SOA yields between low vs. high NOx conditions in the presence of high SO2. The manuscript needs to be thoroughly revised to accurately reflect what the data are showing. If one set of data is showing one thing and another set of data is showing the opposite, the authors need to discuss both datasets equally and cannot conclude that SO2 effect can compensate NOx effect.

The authors concluded that the suppression effect of NOx on SOA yields is mainly due to suppression of nucleation (absence of particle surface area as condensation sink) rather than decrease of condensable materials. If particle surface area plays a role, this will point to the importance of loss process of oxidation products via chemical reactions and/or chamber wall loss. However, the effect of loss of organic vapors on chamber walls is not considered in this study. Nevertheless, previous studies on a-pinene oxidation suggested that SOA yield is independent of particle surface area. In this regard, the interpretation that the suppression effect of NOx arises from a lack of particle surface area appears to be at odds with previous studies. All in all, it is not clear how the absence of particle surface area can explain the suppressed SOA yields under high NOx condition in this study.

The authors explained the effect of SO2 as 1) inducing new particle formation and providing surface area for vapor condensation, 2) acid-catalyzed particle-phase reactions. I have the same question regarding the first explanation, i.e., what is the role of vapor wall loss (if any), and how does one reconcile this explanation with findings from previous studies? Also, what is the effect of SO2 on gas-phase chemistry and SOA yield? This is not considered.

It appears that the SOA yields in high SO2 experiments might be overestimated by double counting the density of ammonium sulfate/ammonium bisulfate in the SOA mass calculation. This is not entirely clear.

Finally, the authors need to conduct a more careful and accurate comparison with previous studies. It was noted that in high SO2 conditions, their findings that SOA yields are comparable under high NOx and low NOx conditions are in line with Sarrafzadeh et al. and Eddingsaas et al. I do not think that the data in Eddingsaas et al. showed this. SOA yields are also a function of deltaMo (as well as various experimental conditions and parameters) and this could play a role, see detailed comment below. Also, the a-pinene yields in this study under comparable NOx/SO2/OH exposure are much lower than Eddingsaas et al.. This is not mentioned and discussed in the manuscript.

Major revisions are needed before the manuscript can be published. Specific comments are listed below.

Detailed comments

1. Line 18-20. This statement is not true for limonene data presented in this study.

2. Line 79 -81. This sentence seems to imply that previous studies that used higher NOx and SO2 concentrations are not atmospherically relevant. I think these sentences should be revised and clarified to more accurately reflect the experimental design and results from previous studies. For instance, the use of high levels of NOx (e.g., from HONO or CH3ONO) in some studies is to push the RO2 radical fate to the extreme (i.e., RO2+NO or RO2+NO2) to investigate SOA yields and composition under such conditions. Thus, the use of high levels of NOx do not necessarily mean that the results are not applicable to ambient conditions.

3. Line 132. Is the organic aerosol density 1.32 g/cm3 from Eddingsaas et al. (2012a)?

If this is the case, note that this density used in Eddingsaas et al. is directly taken from the results in Ng et al. (2007), and that this density was obtained in the presence of seeds already. Therefore, it appears that there might be a double counting of the density of ammonium sulfate/ammonium bisulfate in the data presented here?

4. Line 162. How were OH and O3 formed in the experiments (under each combination of NOx/SO2 condition). Please provide more info. Also, please provide typical time profiles of VOC (either a-pinene of limonene), O3, OH, NO, NO2, SO2 for each combination of NOx/SO2 condition. These are important to help the readers obtain a better idea of the reaction pathways/regimes under each condition.

5. Line 163. Was all the VOC reacted in each experiment?

6. Lin 163. There is no "typical" experiment in this study, as each experiment was conducted under a different NOx/SO2 condition. Please clearly state that this is only for low NOx condition. Also, what about high NOx condition? Was it exclusively OH reaction? Please also specify clearly.

7. Line 173 – 177. Here, under low NOx condition, RO2+NO dominates throughout the entire experiment (RO2+HO2 only contributes to ∼40% at most).

a. These sentences clearly demonstrate the shortcomings of classifying the experiments as low NOx vs. high NOx as discussed in Wennberg et al. (IGAC news, 2103). I suggest the authors to characterize reactions conditions by explicitly stating the RO2 fates, rather than as low vs. high NOx.

b. It is stated that under low NOx conditions, in the beginning of the experiment, a trace amount of NO is formed from photolysis of HONO from the chamber wall. Is this just in the beginning of the experiment, or there is a continuous NO source from HONO photolysis throughout the entire experiment? Please specify.

8. Line 189. The authors attributed the lower particle number concentration and nucleation rate at high NOx to the decreasing RO2+RO2 reaction in the presence of NOx.

However, in line 182, the authors noted that RO2+RO2 reaction is negligible in this study to start with. Please reconcile these seemingly contradictory statements. Also, can be suppressed nucleation under high NOx due to the higher volatility of organic nitrates as compared to peroxides (from RO2+HO2)?

9. Line 205-206. There is nucleation (from organics) in the presence of NOx as shown in Fig. 4. In this sense, "absence of nucleation" here is a bit confusing. Perhaps would be clearer to say "absence of seed particles".

10. Line 211 and Figure 3. The author concluded that the suppression effect of NOx on SOA yields was mainly due to suppression of nucleation, i.e., to the absence of particle surface as condensation sink. Many critical aspects are not discussed, making this conclusion not well-justified and well-supported.

a. If the absence of seed particle surface area is the reason for the low yield under high NOx condition (at low SO2), this will point to the importance of loss of semivolatile species via chemical reactions or chamber wall loss (Kroll et al., ES&T, 2007). However, the effect of vapor wall loss in not considered in this study. Zhang et al. (PNAS, 2014) first systematically investigated the effects of particle surface area and vapor wall loss on SOA yields. For a-pinene photooxidation and ozonolysis specifically, it was found that SOA yields are largely independent of seed surface area (McVay et al., 2016, Nah et al., ACP, 2016; Nah et al., ACP, 2017). Therefore, taken all these together, it is not clear how the absence of particle surface area can explain the suppressed SOA yields under high NOx condition in this study.

b. The authors dismissed the "decrease of condensable organic materials" in high NOx conditions as an explanation for the observed decrease in yield. Why? If more volatile organic nitrates are formed in high NOx conditions, why can't this be an explanation for the suppressed SOA yield? For limonene data (Line 213-218), the authors appeared to embrace the role of volatility of oxidation products.

c. Line 217. How does the different range in VOC/NOx for a-pinene and limonene

experiments explain the differences in yields in high SO2 conditions? Please elaborate and explain clearly.

11. Line 225-237. Comparisons with previous studies. Many critical details are not considered and discussed. I think the authors jumped to the conclusion on whether their study agree/disagree with previous studies too quickly.

a. Line 225. This sentence is only true for a-pinene data in this study, but not for limonene. Please state clearly.

b. Seed particles were generated via SO2 oxidation in this study (for high SO2 experiments). Previous studies directly injected seeds into the chamber. In comparing SOA yields, the author should also consider the role of gas-phase chemistry and particle-phase chemistry. For instance, what about the reaction of SO2 and criegee intermediates? What about the effect of particle acidity on particle-phase reactions (in this study vs. previous studies)? Please discuss.

c. The experiments in this study were conducted in the presence of humidity but previous studies were mostly conducted under dry conditions. RH can affect gas-phase and particle-phase chemistry, and subsequently SOA yields.

d. The authors noted that the finding that SOA yields at high NOx is comparable to that at low NOx in high SO2 conditions is in line with findings in Sarrafzadeh et al. (2016) and Eddingsaas et al. (2012a). I do not think that the data in Eddingsaas et al. showed that "in presence of seed aerosol, the difference in the SOA yield between low and high NOx is much reduced". SOA yield is also a function of deltaMo. Considering the data in Table 1 of Eddingsaas et al., the difference in yields between low and high NO experiments for nucleation is 19%, and for seeded experiments are 15% and 10%. However, the difference in deltaMo for the nucleation experiments is also the largest and this will play a role in the yield difference.

e. The SOA yields in this study are much lower than previous studies, why? Considering the low NOx low SO2 experiment, with OH dose of 1e11 molecules cm-3 s, the yield in this study is 7%. However, the corresponding yield in Eddingsaas et al. is > 30% (Figure 3 of Eddingsaas et al.).

12. Line 238 onwards, effect of SO2.

a. One of the proposed reasons to explain the effect of SO2 is that it induces nucleation and provides more particle surface area for condensation. Again, if this is the case, it will point to the importance of loss of organic vapors to chamber walls, though previous studies suggested that this process does not effect SOA yields from a-pinene oxidations to a large extent. With this, it is not clear if this is indeed a reason for the observed SO2 effect. Please explain.

b. Line 258. Is "counterbalance" the appropriate word? If the suppression effect of NOx is counterbalanced by the enhancement effect SO2, in going from "low NOx low SO2" to "high NOx high SO2" one shall not observe change in SOA yields? Also, note that the limonene data showed very different trends comparing to the a-pinene data. This needs to be accurately and clearly stated.

13. Line 315-318. This explanation is a stretch and not well-justified. There is extensive fragmentation in the AMS and so the H/C ratios of oxidation product molecules do not necessarily translate to the H/C ratios measured. As shown in Chhabra et al., not all experiments conducted under low NOx condition have higher H/C ratios.

14. Line 262 onwards. Did the ratio of nitrate mass concentration to organics mass change over time?

Minor comments

1. Line 72. Why "in contrast"? 2. Line 84 citation. There are more studies on OH oxidations of a-pinene and they should also be cited here (for example, some of the studies cited in page 2). 3. Line 125. "mass" should be "volume"? SMPS measures volume concentration. 4. Line 126. Delete "with". 5. Line 256. Sentence not clear. 6.

Figure 1 caption should specify the SO2 condition.

[Figure]

---

## Author Comment (AC1) · 28 Aug 2017

**Responses to Referee # 1**

We thank the reviewer for the careful review of our manuscript. The comments and suggestions are greatly appreciated. All the comments have been addressed. In the following, please find our responses to the comments one by one and the corresponding revisions made to the manuscript. The original comments are shown in italics. The revised parts of the manuscript are highlighted.

**Anonymous Referee #1**

*This manuscript describes the competing roles of NOx and SO2 on SOA formation of a-pinene and*

*limonene. The ability of SO2 to enhance seed aerosol surface area appears to be a dominant factor,*

*and that enhancing seed aerosol reduces the NOx suppression of SOA yields, at least in some*

*monoterpenes. The authors use their AMS data to determine the role of organic nitrates in SOA, and*

*find that organic nitrates account for a substantial fraction of the SOA mass. Overall, this is an*

*interesting piece of work, and warrants publication in the ACP following some revision.*

*Major Comments*

*The nature of the experimental design was not so much to look at the impact of SO2 – but to look at*

*the role of a sulfate seed aerosol. From the manuscript, my interpretation is that the SO2 additions*

*were used to nucleate (inorganic) seed aerosol. Was there any SO2 left over to impact VOC oxidation?*

*It is not clear to me if the SO2 additions really paralleled the NOx additions, because the*

*experimental design was different. That's not to say that these aren't valuable experiments that add to*

*the literature! I merely question whether this was truly an 'SO2 addition' rather than a 'sulfate*

*aerosol addition' to VOC oxidation experiments.*

**Response:**

We thank the reviewer for the supporting remarks.

In the experiments with $SO_2$ added, $SO_2$ concentration decreased slowly and most of $SO_2$ was still left (typically around 8 ppb) at the end of an experiment because of the low reactivity of $SO_2$ with OH

($\sim 2\times10^{-12}$ molecules$^{-1}$ cm$^3$ s$^{-1}$ at 298 K). $SO_2$ time series in a typical experiment are shown in a newly added figure (Fig. S2). Therefore, the experiments with $SO_2$ not only included the effect of sulfate formed from $SO_2$ oxidation as seed but also the potential role of $SO_2$ on VOC oxidation, although the role on VOC oxidation turned out to  be likely not significant.  We have added the follow sentence in the revised manuscript to clarify this point.

"$SO_2$ concentration decayed slowly in the experiments with $SO_2$ added and most of the $SO_2$ was still left (typically around 8 ppb from initial 15 ppb) at the end of an experiment due to its low reactivity with OH.  Typical $SO_2$ time series in high $SO_2$ experiments are shown in Fig S2."

Although the $SO_2$ addition did not exactly parallel $NO_x$ addition, by adding $SO_2$ and inducing
nucleation first, we can make sure that in high $SO_2$ conditions enough nucleated particles were
represent for the oxidation products to condense on once VOC oxidation started. Otherwise, it would
be unclear whether the low SOA yield at high $NO_x$ was due to missing nucleation or lack of
condensable products. In this way, we can somewhat differentiate the role of promoting nucleation
from the role of affecting the condensable products from VOC oxidation.

In the revised manuscript, we have added the following sentence.

"Adding $SO_2$ first and initializing nucleation by $SO_2$ photooxidation ensured that enough nucleating
particles were present when VOC oxidation started."

The effect of $SO_2$ on VOC oxidation is provided in the response to one similar comment below (Pg. 7,
lines 205-209).

*SOA yield is influenced strongly by OA mass. The authors plot SOA yield versus OH dose, which is*
*certainly a useful figure to see – but it is hard to compare the SOA yields if the SOA mass has not*
*been accounted for. The authors need to also show SOA yield versus OA mass so that the readers can*
*contrast the relationships to other studies. It would be useful to compare the SOA yields to other*
*studies: how do the yield values compare to other measurements of OH oxidation of a-pinene? This*
*will allow readers to place the studies in context.*

**Response:**

We have accepted the reviewer's suggestion. In the revised manuscript, we have added a figure of
SOA yield versus OA mass concentration (Fig. S8 and S9) and compared the SOA yield in this study
to previous studies. We have also discussed other factors influencing SOA yield.

*The results of SO2 and NOx effects on SOA yield are consistent with the Sarrafzadeh and Eddingsaas*
*studies, which found that the presence of seed aerosol suppresses the 'NOx effect' on SOA yield.*
*However, they contradict previous studies (e.g. Ng et al. 2007, Presto et al. 2005). The authors need*
*to do a better job of contrasting their studies – they attribute the difference to a vague collection of*
*parameters (e.g. NO:NO2 ratio, OH concentrations, etc.). It would be extremely helpful if the authors*
*could synthesize the information (i.e. put numbers on those parameters) to help readers understand*
*the differences in experimental conditions across the studies. A table would be particularly helpful.*

**Response:**

We have accepted the reviewer's suggestions. In the revised manuscript, we have added one table
summarizing the reaction conditions of previous studies (Table. S2) and elaborated the discussion
related to the difference between our study and previous studies.

"…The reaction conditions of this study often differ from those described in the literature (see Table S2).

The difference in these conditions can result in both different apparent dependence on specific parameters and the varied SOA yield. For example, SOA yield from α-pinene photooxidation at low $NO_x$ in this study appeared to be much lower than that in Eddingsaas et al. (2012a). The difference between the SOA yield in this study and some of previous studies and between the values in the literature can be attributed to several reasons: 1) $RO_2$ fates may be different. For example, in our study at low $NO_x$, $RO_2$+NO account for a large fraction of $RO_2$ loss while in Eddingsaas et al. (2012a) $RO_2$+$HO_2$ is the dominant pathway of $RO_2$ loss. This difference in $RO_2$ fates may affect oxidation products distribution. 2) The organic aerosol loading of this study is much lower than that some of previous studies (e.g., Eddingsaas et al. (2012a)) (see Fig. S9). SOA yields in this study were also plotted versus organic aerosol loading to better compare with previous studies (Fig. S8 and S9). 3) The total particle surface area in this study may also differ from previous studies, which may influence the apparent SOA yield due to vapor wall loss (the total particle surface area is often not reported in many previous studies to compare with). 4) RH of this study is different from many previous studies, which often used very low RH (<10%). It is important to emphasize that reaction conditions including the $NO_x$ as well as $SO_2$ concentration range and RH in this study were chosen to be relevant to the anthropogenic-biogenic interactions in the ambient atmosphere. In addition, difference in the organic aerosol density used in yield calculation should be taken into account. In this study, SOA yield was derived using a density of 1 g cm$^{-3}$ to better compare with many previous studies (e.g., (Henry et al., 2012)), while in some other studies SOA yield was derived using different density (e.g., 1.32 g cm$^{-3}$ in Eddingsaas et al. (2012a))."

*Lines 118: the use of the HR-ToF-AMS to derive elemental ratios uses the older Aiken method. However, as the authors note, the newer 2015 approach corrects some underestimation. Because readers may wish to compare results across studies in the future, it is appropriate and prudent to update the results to the newer calculations.*

**Response:**

We have calculated the H/C and O/C data using the newer approach by Canagaratna et al. (2015) and compared them with the data derived from the older method (Aiken et al., 2007) (Fig. S1). The H/C derived using the newer method strongly correlated with that derived using older method and just increased by 27%. Similarly, O/C just increased by 11%. In the revised manuscript, we have discussed this difference.

"The H/C and O/C were also derived using the newer approach by Canagaratna et al. (2015) and compared with the data derived from the Aiken et al. (2007) method. The H/C values derived using the Canagaratna et al. (2015) method strongly correlated with the values derived using Aiken et al.

(2007) method (Fig. S1) and just increased by 27% as suggested by Canagaratna et al. (2015). Similar results were found for O/C and there was just a difference of 11% in O/C. Since only relative difference in elemental composition of SOA is studied here, only the data derived using Aiken et al.

(2007) method are shown as the conclusion was not affected by the methods chosen."

*Line 128: the authors note that they account for particle wall loss and dilution loss, but not for vapor*

*wall loss. Recent papers have shown this to be a chemically-dependent and substantial effect on SOA*

*yields, and most rigorous SOA yield work now accounts for these effects. How will ignoring vapor*

*wall loss influence the results – and the interpretation thereof?*

**Response:**

The wall loss of vapors causes an under-estimate of the SOA yield. In the revised manuscript, we have estimated the influence of the vapor wall loss on SOA yield using the measured wall loss rate of vapors. And we have added a section to address the influence of vapor wall loss.

**"Wall loss of organic vapors**

The loss of organic vapors on chamber walls can influence SOA yield (Kroll et al., 2007;

Zhang et al., 2014; Ehn et al., 2014; Sarrafzadeh et al., 2016; McVay et al., 2016; Nah et al., 2016;

Matsunaga and Ziemann, 2010; Ye et al., 2016; Loza et al., 2010). The wall loss rate of organic vapors in our chamber was estimated by following the decay of organic vapor concentrations after photooxidation was stopped in the experiments with low particle surface area ($\sim5\times10^{-8}$ cm$^2$ cm$^{-3}$) and thus low condensational sink on particles. Such method is similar to the method used in previous studies (Ehn et al., 2014; Sarrafzadeh et al., 2016; Krechmer et al., 2016; Zhang et al., 2015). A high- resolution time-of-flight chemical ionization mass spectrometer (HR-ToF-CIMS, Aerodyne Research

Inc.) with nitrate ion source ($^{15}NO_3^-$) was used to measure semi/low-volatile organic vapors. The details of the instrument were described in our previous studies (Ehn et al., 2014; Sarrafzadeh et al.,

2016). The decay of vapors started from the time when the roof of the chamber was closed. The data were acquired at a time resolution of 4 s. A typical decay of low-volatile organics is shown in Fig. S3

and the first-order wall loss rate was determined to be around $6\times10^{-4}$ s$^{-1}$.

The SOA yield was not directly corrected for the vapor wall loss, but the influence of vapor wall loss on SOA yield was estimated using the method in the study of Sarrafzadeh et al. (2016) and the details of the method are described therein. Briefly, particle surface and chamber walls competed for the vapor loss (condensation) and the condensation on particles led to particle growth. The fraction of organic vapor loss to particles in the sum of the vapor loss to chamber walls and to particles ($F_p$)

was calculated. The vapor loss to chamber walls was derived using the wall loss rate. The vapor loss to particles was derived using particle surface area concentration, molecular velocity and an accommodation coefficient $\alpha_p$ (Sarrafzadeh et al., 2016). $1/F_p$ ($f_{corr}$) provides the correction factor to obtain the "real" SOA yield. $f_{corr}$ is a function of particle surface area concentration and
accommodation coefficient as shown in Fig. S4. Here a range of 0.1-1 for $\alpha_p$ was used, which is
generally in line with the ranges of $\alpha_p$ found by Nah et al. (2016) by fitting a vapor-particle dynamic
model to experimental data. At a given $\alpha_p$, the higher particle surface area, the lower $f_{corr}$ and the
lower the influence of vapor wall loss are because most vapors condense on particle surface and vice
versa. At a given particle surface area, $f_{corr}$ decreases with $\alpha_p$ because at higher $\alpha_p$ a larger fraction of
vapors condenses on particles. An average molecular weight of 200 g/mol was used to estimate the
influence of vapor wall loss. For the aerosol surface area range in most of the experiments in this
study, the influence of vapor wall loss on SOA yield was relatively small (<~40% for particle surface
area larger than $3 \times 10^{-6}$ cm$^2$ cm$^{-3}$, Fig. S4). Yet, for the experiments at high NO$_x$ and low SO$_2$ for $\alpha$-
pinene and limonene, the influence of vapor wall loss on SOA can be high due to the low particle
surface area, especially at lower $\alpha_p$."

*In the Introduction, the authors do a good job of summarizing the reasons why such a study would be*
*interesting. Much of the discussion focuses on the role of NOx on SOA yields – this is reasonable as*
*most of the literature has focused on that problem! However, there is some relatively recent literature*
*regarding the role of SO2 in affecting SOA chemistry and monoterpene OH oxidation that the authors*
*should consider. In particular:*

*Photooxidation of cyclohexene in the presence of SO2: SOA yield and chemical composition.*

*Shijie Liu, Long Jia, Yongfu Xu, Narcisse T. Tsona, Shuangshuang Ge, and Lin Du. Atmos. Chem.*
*Phys. Discuss., doi:10.5194/acp-2017-30, 2017*

*Synergetic formation of secondary inorganic and organic aerosol: effect of SO2 and NH3 on particle*
*formation and growth. Biwu Chu, Xiao Zhang, Yongchun Liu, Hong He, Yele Sun, Jingkun Jiang,*
*Junhua Li, and Jiming Hao. Atmos. Chem. Phys., 16, 14219-14230, doi:10.5194/acp-16-14219-2016,*
*2016*

*Formation of secondary aerosols from gasoline vehicle exhaust when mixing with SO2. T. Liu, X.*
*Wang, Q. Hu, W. Deng, Y. Zhang, X. Ding, X. Fu, F. Bernard, Z. Zhang, S. Lü, Q. He, X. Bi, J. Chen,*
*Y. Sun, J. Yu, P. Peng, G. Sheng, and J. Fu. Atmos. Chem. Phys., 16, 675-689, doi:10.5194/acp-16-*
*675-2016, 2016*

*Anthropogenic Sulfur Perturbations on Biogenic Oxidation: SO2 Additions Impact Gas- Phase OH*
*Oxidation Products of _- and _-Pinene. Beth Friedman, Patrick Brophy, William H. Brune, and*
*Delphine K. Farmer. Environmental Science & Technology 2016 50 (3), 1269-1279. DOI:*
*10.1021/acs.est.5b05010*

*Is there any evidence for organic sulfates in the SOA from the AMS data? This has been a subject of*

*some debate in the literature, and an additional datapoint would be useful. This may also clarify the*

*role of acid catalysis, as I believe that has been linked to the formation of organic sulfates.*

**Response:**

We thank the reviewer for raising these papers. In the revised manuscript, we have enriched the discussion on the role of $SO_2$ by including some of the papers.

From our AMS data, we did not find evidence of organic sulfate. For SOA formed at high $SO_2$, we found no significant organic fragments containing sulfur. Also the fragment $CH_3SO_2^+$ from organic sulfate suggested by Farmer et al. (2010) was not detectable in our data. We found that the pattern of sulfate in mass spectra had no significant difference from the pattern of pure ammonium sulfate.

However, we would like to note that AMS has very limited capability to differentiate organic sulfate and inorganic sulfate (Farmer et al., 2010).

Moreover, according to the literature, organic sulfate is mainly formed by aqueous reaction of sulfate with organics. In the conditions of our study, there was no aqueous phase as we stated based on the

AIM model. Therefore, experimental conditions in our study did not favor the formation organic sulfate.

In the revised manuscript, we have clarified these results.

"In addition, from the AMS data of SOA formed at high $SO_2$ no significant organic fragments containing sulfur were found. Also the fragment $CH_3SO_2^+$ from organic sulfate suggested by Farmer et al. (2010) was not detected in our data. The absence of organic sulfate tracers is likely due to the lack of aqueous phase in aerosol particles in this study. Therefore, the influence of $SO_2$ on gas phase chemistry of organics and further on SOA yield via affecting gas phase chemistry is not important in this study."

Minor Comments

*Line 136. The authors note an average RH of 28-42% for the experiments. This seems like a relatively*

*large range: will this affect the SOA yields, or interpretation of the data?*

**Response:**

The average RH was in the range of 28-42% taking into account all experiments. Actually, except one experiment, the average RH was in the range of 28-34%. For particle phase reactions, the particle water content absorbed by organic aerosol in the range of 28-42% RH is low and the difference of water content between 28% and 42 % is very minor (typically <~2% of the particle volume based on our hygroscopic growth measurement). The RH variations are not expected to significantly change the particle phase chemistry. Moreover, since water vapor is abundant and in excess in the gas phase, the

RH variations are not expected to significant change gas phase chemistry either. Therefore, we do not
expect that would significantly change the SOA yield. In the revised manuscript, we have clarified
this point.

"The average RH for the period of monoterpene photooxidation was 28-34% except for one
experiment with average RH of 42% RH."

*Re: Discussion of SO2 effects. The authors dominantly attribute the enhancement of SOA by SO2 to*
*increased particle surface area, or perhaps to acid catalysis. These seem like extremely likely reasons;*
*however, there is one study that suggests that SO2 will influence gas-phase oxidation products*
*(Friedman et al.), which could also be a confounding factor unless all of the SO2 is in the particle*
*phase before VOC oxidation commences… This would be a useful clarification.*

**Response:**

The influence of $SO_2$ on gas phase oxidation is likely to be trivial in this study for two reasons. Firstly,
the reactivity of $SO_2$ with OH is very low ($2\times10^{-12}$ vs. $5.3\times10^{-11}$ molecules$^{-1}$ cm$^3$ s$^{-1}$ for α-pinene with
OH) and $SO_2$ only accounts for a very small fraction of the OH loss (typically ~2% in the beginning of
an experiment). Secondly, the OH concentration is 2-3 orders of magnitude lower than those in the
PAM chamber used by Friedman et al. (2016). Therefore, either the change in OH/HO$_2$ ratio or SO$_3$
concentration, which is attributed to the reason of changed oxidation products by Friedman et al.
(2016), is much lower in our experiments. In the revised manuscript, we have added a brief discussion
of the effect of $SO_2$ on gas phase oxidation of monoterpenes in this study as follows.

"$SO_2$ has been proposed to also affect gas phase chemistry of organics by changing the HO$_2$/OH or
forming SO$_3$ (Friedman et al., 2016). In this study, the effect of $SO_2$ on gas phase chemistry of
organics was not significant because of the much lower reactivity of $SO_2$ with OH compared with α-
pinene and limonene (Atkinson et al., 2004, 2006; Atkinson and Arey, 2003) and the low OH
concentrations (2-3 orders of magnitude lower than those in the study by Friedman et al. (2016)).
Moreover, reactions of RO$_2$ with $SO_2$ was also not important because the reaction rate constant is very
low ($<10^{-14}$ molecule$^{-1}$ cm$^3$ s$^{-1}$) (Lightfoot et al., 1992; Berndt et al., 2015). In addition, from the AMS
data of SOA formed at high $SO_2$ no significant organic fragments containing sulfur were found. Also
the fragment CH$_3$SO$_2^+$ from organic sulfate suggested by Farmer et al. (2010) was not detected in our
data. The absence of organic sulfate tracers is likely due to the lack of aqueous phase in aerosol
particles in this study. Therefore, the influence of $SO_2$ on gas phase chemistry of organics and further
on SOA yield via affecting gas phase chemistry is not important in this study."

*Technical comments.*

*Line 26: should read "compared to low NOx"*

**Response:** Corrected.

*Line 29: should read "SO2 can compensate for such effects"*

**Response:** Corrected.

*Introduction: line 34: sentence has repetitive 'importants': consider removing at least one (e.g. "SOA*
*is an important class of atmospheric aerosol" seems like an unnecessary statement for the journal's*
*audience). This adjective is used heavily throughout the introduction (lines 45, 49), and I recommend*
*removing or replacing the adjective to improve readability.*

**Response:** We have accepted the reviewer's suggestion. In the revised manuscript, we have removed
the "as an important class of atmospheric aerosol", and removed or replaced "important" where it is
necessary.

*Line 56: hydroperoxides should be plural*

**Response:** Corrected.

*Line 57: need comma between 'NO' and 'forming'*

**Response:** Corrected.

*Line 87: should read "might have either counteracting or synergistic effects on SOA: : :"*

**Response:** Corrected.

*Line 126: remove the with following 'multiplied by'*

**Response:** Done.

Line 135: should read 'there was no aqueous..'

**Response:** Corrected.

*Line 221, remove comma between 'that' and 'high'*

**Response:** Done.

*Line 360: should read 'in the ambient atmosphere'*

**Response:** Corrected.

**References**

[revised manuscript text omitted]

---

## Author Comment (AC2) · 28 Aug 2017

**Responses to Referee # 2**

We thank the reviewer for carefully reviewing our manuscript; the comments and suggestions are very helpful and greatly appreciated. All the comments have been addressed. We believe that revisions based on these comment have substantially improved our manuscript. In the following please find our responses to the comments one by one and the corresponding changes made to the manuscript. The original comments are shown in italics. The revised parts of the manuscript are highlighted.

Before we start our responses to the reviewer, we would like to briefly clarify the motivation of this manuscript. Our primary goal of this study is to investigate how and to what extent small ambient inorganic trace gases, here $NO_x$ and $SO_2$, affect the SOA formation in the ambient anthropogenic-biogenic interactions. We aimed to study the more complex situation found in the ambient atmosphere instead of pure VOC reaction systems. Our main goal is not so much to provide a single absolute value of SOA yield to modelers nor to simply "improve" values of the SOA yields from previous studies because all chamber studies have operational limitations compared to the atmosphere (e.g. wall effects). Often different studies have distinct physical and chemical regimes due to the different operational limitations and diverse experimental conditions e.g. chamber size, radical generation, and photolysis rates. We as well as the community are in the course of addressing the influence of wall loss of vapors. We believe that, rather denying the findings from previous studies, our study provided additional knowledge and insights to existing understanding of SOA formation in certain conditions of the real atmosphere. This is based on the fact that our experiments were conducted under conditions relatively close to the ambient anthropogenic-biogenic interactions, including ambient RH, concentrations of $SO_2$, $NO_x$, and VOC close to ambient levels, natural sunlight, and low surface-to-volume ratio of our large chamber.

*Anonymous Referee #2*

*This chamber study investigated the effects of SO2 and NOx (NO) on SOA formation from photooxidation of a-pinene and limonene. It was found that SO2 enhanced SOA yield while NOx suppressed SOA yield. The suppression effect of NOx was attributed to the suppressed new particle formation and thus a lack of particle surface area for organics to condense on. The authors concluded that SO2 oxidation produced high number of particles and compensated for the suppression of SOA yield by NOx. SOA composition measured by AMS was also presented and discussed.*
*This is an interesting study. The gas- and particle-phase measurements are comprehensive and include several important species that have not been typically characterized in previous studies (e.g., OH, HO2 and RO2). The experiments appeared to be carefully conducted. However, I have major concerns regarding data interpretation and some conclusions in the manuscript.*
*One of the central themes of the manuscript is that the suppression effect of NOx on SOA formation*

*can be compensated by the presence of SO2. This conclusion is not accurate based on all the data*

*presented in this manuscript. For a-pinene, it appears that under high SO2 conditions, the SOA yields*

*under low vs. high NOx conditions are comparable. However, this is not the case for limonene, where*

*there is still a large difference in SOA yields between low vs. high NOx conditions in the presence of*

*high SO2. The manuscript needs to be thoroughly revised to accurately reflect what the data are*

*showing. If one set of data is showing one thing and another set of data is showing the opposite, the*

*authors need to discuss both datasets equally and cannot conclude that SO2 effect can compensate*

*NOx effect.*

**Response:**

A key result in our study is that for both α-pinene and limonene, the difference in SOA yield between high $NO_x$ and low $NO_x$ was much reduced in the presence of $SO_2$, although for limonene the SOA

yield at high $NO_x$ and high $SO_2$ was still lower than the yield at low $NO_x$. This result indicates that the suppression of SOA yield by $NO_x$ was compensated to a large extent by $SO_2$. This conclusion holds regardless the difference in the detailed results between α-pinene and limonene system.

In the revised manuscript, we have revised the conclusion to better represent the results from both

α-pinene and limonene cases. The following sentence has been revised in the conclusion part:

"$SO_2$ oxidation produced high number concentration of particles and compensated for the suppression of SOA yield by $NO_x$ to a large extent."

The abstract has been revised accordingly as follows.

"However, in the presence of $SO_2$ which induced high number concentration of particles after oxidation to $H_2SO_4$, the suppression of the mass yield of SOA by $NO_x$ was completely or partly compensated."

*The authors concluded that the suppression effect of NOx on SOA yields is mainly due to suppression*

*of nucleation (absence of particle surface area as condensation sink) rather than decrease of*

*condensable materials. If particle surface area plays a role, this will point to the importance of loss*

*process of oxidation products via chemical reactions and/or chamber wall loss. However, the effect of*

*loss of organic vapors on chamber walls is not considered in this study. Nevertheless, previous studies*

*on a-pinene oxidation suggested that SOA yield is independent of particle surface area. In this regard,*

*the interpretation that the suppression effect of NOx arises from a lack of particle surface area*

*appears to be at odds with previous studies. All in all, it is not clear how the absence of particle*

*surface area can explain the suppressed SOA yields under high NOx condition in this study.*

**Response:**

In general, condensable SOA materials are chemically produced in the gas-phase and nucleation and condensation on particle surface constitute sinks for the SOA materials besides other sinks e.g. wall loss and flush out. In the case of sufficient nucleation or seeded experiments, condensation on particles is the dominant sink. In absence of nucleation and surface, the other sinks such as wall loss dominate the losses of SOA materials. Therefore, no SOA would be formed without nucleation. This

*a priori* tells nothing about the importance of wall losses when SOA is formed.

The loss of organic vapors to chamber walls can be important for SOA yield, although in our manuscript we had not corrected for it because it is a challenge to quantify it. In the revised manuscript, we have added a section to estimate the influence of vapor wall loss on SOA yield (Sect.

"2.3 Wall loss of organic vapors") and provided more discussion on vapor wall loss. We found that the influence of vapor wall loss on SOA yield is likely to be significant when surface area concentrations of SOA formed were low in the high $NO_x$ and low $SO_2$ conditions. Yet, the influence of vapor wall loss is likely to be not significant at the higher surface area concentrations of SOA

formed in the low $NO_x$ conditions or high $NO_x$ and high $SO_2$ conditions.

We respectfully disagree with the reviewer's statement that "previous studies on α-pinene oxidation suggested that SOA yield is independent of particle surface area" without considering each specific study. Although some studies showed that SOA yield from α-pinene oxidation is independent of seed particle surface area (McVay et al., 2016; Nah et al., 2016), a number of studies showed that SOA

yield from α-pinene ozonolysis or photooxidation depends on particle surface area. For example, our previous studies clearly showed that SOA yields from α-pinene photooxidation depend on surface area (Sarrafzadeh et al., 2016; Ehn et al., 2014), and Eddingsaas et al. (2012) also showed that in

"high NO" conditions, SOA yield from α-pinene photooxidation is much higher with neutral seed than that without seed. The discrepancy in the dependence of SOA yield on particle surface areas in the literature can be attributed to reaction conditions, surface area range and chamber setup. For example, if the reaction produces enough new particles by itself and results in fast particle growth and larger aerosol surface area as the dominant condensational sink for vapors compared to the loss on chamber walls, SOA yield would be less affected by the seed surface area. On the contrary, if VOC

oxidation does not induce nucleation by itself, all vapors would be lost onto the chamber walls and

SOA yield would be essentially zero.

In this study, at high $NO_x$ and low $SO_2$, the particle number and surface area concentrations were low (peak surface area concentration of $6.8\times10^{-7}$ $cm^2$ $cm^{-3}$ and particle-to-chamber surface ratio of

$7.7\times10^{-5}$ for α-pinene), much lower than the aerosol surface area range in the studies by Nah et al.

(2016) and McVay et al. (2016) ($\sim10^{-5}$ $cm^2$ $cm^{-3}$ and particle-to-chamber surface ratio of $>4\times10^{-5}$). In addition, not only seed particle surface area but also total particle surface area formed during reaction provide condensational sinks to compete with vapor wall loss. At such low particle surface area concentrations, the condensation of vapors on particles had a much longer time scale than that of the wall loss and a large fraction of vapors condensed on chamber walls. Therefore, SOA yield was significantly suppressed due to lack of particle surface area.

In the revised manuscript, we have discussed the findings in the literature.

"Artificially added seed aerosol has been shown to enhance SOA formation from α-pinene and

β-pinene oxidation (Ehn et al., 2014; Sarrafzadeh et al., 2016; Eddingsaas et al., 2012a). In some other studies, it was found that the SOA yield from α-pinene oxidation is independent of initial seed surface area (McVay et al., 2016; Nah et al., 2016). The difference in the literature may be due to the range of total surface area of particles, reaction conditions and chamber setup. For example, the peak particle-to-chamber surface ratio for α-pinene photooxidation in this study was $7.7 \times 10^{-5}$ at high $NO_x$

and low $SO_2$, much lower than the aerosol surface area range in the studies by Nah et al. (2016) and

McVay et al. (2016). A lower particle-to-chamber surface ratio can lead to a larger fraction of organics lost on chamber walls. Hence, providing additional particle surface by adding seed particles can increase the condensation of organics on particles and thus increase SOA yield. However, once the surface area is high enough to inhibit condensation of vapors on chamber walls, further enhancement of particle surface will not significantly enhance the yield (Sarrafzadeh et al., 2016)."

*The authors explained the effect of SO2 as 1) inducing new particle formation and providing surface*

*area for vapor condensation, 2) acid-catalyzed particle-phase reactions. I have the same question*

*regarding the first explanation, i.e., what is the role of vapor wall loss (if any), and how does one*

*reconcile this explanation with findings from previous studies? Also, what is the effect of SO2 on*

*gas-phase chemistry and SOA yield? This is not considered.*

**Response:**

The role of vapor wall loss can be referred to our response to the comment above. Wall loss of vapors leads to an underestimate of SOA yield. Condensation of vapors onto aerosol particle surface competes with the loss of vapors on chamber walls. Therefore the surface area provided by nucleation and growth of particles in the presence of $SO_2$ enhanced the SOA yield in this study.

Only few studies have investigated the effect of $SO_2$ on the SOA yield from α-pinene oxidation. More studies investigated the effect of seed aerosol and particle acidity. Kleindienst et al. (2006) attributed the increase of SOA yield in the presence of $SO_2$ to the formation of $H_2SO_4$ acidic aerosol. While particle acidity may contribute to the increased SOA yield, especially at high $NO_x$, the effect of facilitating nucleation and further providing surface area seems to be more important in our study.

The importance of the $SO_2$ via nucleating and providing particle surface depends on the particle surface area in the absence of $SO_2$ because the competition for the condensation of vapors between particles and wall depends on particle surface area. When VOC oxidation does not form enough new particles and particle surface by itself, the role of $SO_2$ via nucleating and providing particle surface in enhancing SOA yield is more important. In the revised manuscript, we have added more discussion on this aspect.

"Particle acidity may also play a role in affecting the SOA yield in the experiments with high $SO_2$.

Particle acidity was found to enhance the SOA yield from α-pinene photooxidation at high $NO_x$

(Offenberg et al., 2009) and "high NO" conditions (Eddingsaas et al., 2012a). Yet, in low $NO_x$

condition, particle acidity was reported to have no significant effect on the SOA yield from α-pinene photooxidation (Eddingsaas et al., 2012a; Han et al., 2016). According to these findings, at low $NO_x$

the enhancement of SOA yield in this study is attributed to the effect of facilitating nucleation and providing more particle surface by $SO_2$ photooxidation. At high $NO_x$, the effect in enhancing new particle formation by $SO_2$ photooxidation seems to be more important, although the effect of particle acidity resulted from $SO_2$ photooxidation may also play a role."

The effect of $SO_2$ on gas phase chemistry is not significant in this study because the reaction rate of

$SO_2$ with OH (~$2\times10^{-12}$ molecules$^{-1}$ cm$^3$ s$^{-1}$) and with $RO_2$ (<$10^{-14}$ molecule$^{-1}$ cm$^3$ s$^{-1}$) are very low (Lightfoot et al., 1992; Berndt et al., 2015). From the AMS data of SOA formed at high $SO_2$, we found no significant organic fragments containing sulfur. Also the fragment $CH_3SO_2^+$ suggested by

Farmer et al. (2010) was not detected in our data. Therefore, we conclude that in our study, the effect of $SO_2$ on gas phase chemistry of organics and thus further on SOA yield via affecting gas phase chemistry is not important.

In the revised manuscript, we have added the following discussion about this point.

"$SO_2$ has been proposed to also affect gas phase chemistry of organics by changing the $HO_2$/OH or forming $SO_3$ (Friedman et al., 2016). In this study, the effect of $SO_2$ on gas phase chemistry of organics was not significant because of the much lower reactivity of $SO_2$ with OH compared with

α-pinene and limonene (Atkinson et al., 2004, 2006; Atkinson and Arey, 2003) and the low OH

concentrations (2-3 orders of magnitude lower than those in the study by Friedman et al. (2016)).

Moreover, reactions of $RO_2$ with $SO_2$ was also not important because the reaction rate constant is very low (<$10^{-14}$ molecule$^{-1}$ cm$^3$ s$^{-1}$) (Lightfoot et al., 1992; Berndt et al., 2015). In addition, from the AMS

data of SOA formed at high $SO_2$ no significant organic fragments containing sulfur were found. Also the fragment $CH_3SO_2^+$ from organic sulfate suggested by Farmer et al. (2010) was not detected in our data. The absence of organic sulfate tracers is likely due to the lack of aqueous phase in aerosol particles in this study. Therefore, the influence of $SO_2$ on gas phase chemistry of organics and further on SOA yield via affecting gas phase chemistry is not important in this study."

*It appears that the SOA yields in high SO2 experiments might be overestimated by double counting*

*the density of ammonium sulfate/ammonium bisulfate in the SOA mass calculation. This is not entirely*

*clear.*

**Response:**

The density of SOA in high $SO_2$ experiments was not double counted. In the revised manuscript, we have clarified this point. Please also refer to our response to the similar comment below ("detailed comment" 3).

"…and their respective density (1.32 g cm$^{-3}$ for organic aerosol from one of our previous studies (Flores et al., 2014) and the literature (Ng et al., 2007) and ~1.77 g cm$^{-3}$ for ammonium sulfate/ammonium bisulfate)…"

We would like to note that the SOA yield in this study was derived by adjusting the density of SOA to

1 g cm$^{-3}$.

*Finally, the authors need to conduct a more careful and accurate comparison with previous studies. It*

*was noted that in high SO2 conditions, their findings that SOA yields are comparable under high NOx*

*and low NOx conditions are in line with Sarrafzadeh et al. and Eddingsaas et al. I do not think that*

*the data in Eddingsaas et al. showed this. SOA yields are also a function of deltaMo (as well as*

*various experimental conditions and parameters) and this could play a role, see detailed comment*

*below. Also, the a-pinene yields in this study under comparable NOx/SO2/OH exposure are much*

*lower than Eddingsaas et al.. This is not mentioned and discussed in the manuscript.*

**Response:**

In the revised manuscript, we have added more discussion to compare this study with previous studies and added a table summarizing previous studies.

Please refer to our detailed responses to the specific comments below ("detailed comments" 11d, e).

*Major revisions are needed before the manuscript can be published. Specific comments are listed*

*below.*

*Detailed comments*

*1. Line 18-20. This statement is not true for limonene data presented in this study.*

**Response:**

In the revised manuscript, we have modified this statement. Please refer to our response to a similar comment above (Pg. 1, lines 35-43).

*2. Line 79 -81. This sentence seems to imply that previous studies that used higher NOx and SO2*

*concentrations are not atmospherically relevant. I think these sentences should be revised and*

*clarified to more accurately reflect the experimental design and results from previous studies. For*

*instance, the use of high levels of NOx (e.g., from HONO or CH3ONO) in some studies is to push the*

*RO2 radical fate to the extreme (i.e., RO2+NO or RO2+NO2) to investigate SOA yields and*

*composition under such conditions. Thus, the use of high levels of NOx do not necessarily mean that*

*the results are not applicable to ambient conditions.*

**Response:**

We understand that in some studies high concentration of $NO_x$ is intentionally used to push the $RO_2$

radical fate to an extreme in order to study the SOA yields and composition. Nevertheless, experiments under ambient levels of high $NO_x$ concentrations are more transferable to the ambient anthropogenic-biogenic interaction than the experiments conducted at extremely high $NO_x$

concentrations considering the shortened lifetime of $RO_2$ and the potential secondary processes as well as the effect of $NO_x$ on OH concentration at extremely high $NO_x$. In the revised manuscript, we have modified this sentence.

"For example, many studies used very high $NO_x$ and $SO_2$ concentrations (up to several hundreds of ppb). High $NO_x$ can make the $RO_2$ radical fate dominated by one single pathway (i.e., $RO_2$+NO or

RO$_2$+NO$_2$) to investigate SOA yields and composition under such conditions. Yet, the effects of NO$_x$

and SO$_2$ at concentration ranges for ambient anthropogenic-biogenic interactions (sub ppb to several tens of ppb for NO$_2$ and SO$_2$) have seldom been directly addressed."

*3. Line 132. Is the organic aerosol density 1.32 g/cm3 from Eddingsaas et al. (2012a)? If this is the*

*case, note that this density used in Eddingsaas et al. is directly taken from the results in Ng et al.*

*(2007), and that this density was obtained in the presence of seeds already. Therefore, it appears that*

*there might be a double counting of the density of ammonium sulfate/ammonium bisulfate in the data*

*presented here?*

**Response:**

Organic aerosol density was based on our previous study (Flores et al., 2014) as well as the study of

Eddingsaas et al. (2012) and thus Ng et al. (2007).

Although the density of organic aerosol in the study of Ng et al. (2007) was obtained in presence of seed, the contribution of seed aerosol (ammonium sulfate) to particle volume and thus density has been taken into account (Bahreini et al., 2005) . The value reported by Ng et al. (2007) is the density of organic aerosol instead of the density of mixed aerosol. There is no double counting of the density of ammonium sulfate/ammonium bisulfate in our study.

4. *Line 162. How were OH and O3 formed in the experiments (under each combination of NOx/SO2*

*condition). Please provide more info. Also, please provide typical time profiles of VOC (either*

*a-pinene of limonene), O3, OH, NO, NO2, SO2 for each combination of NOx/SO2 condition. These*

*are important to help the readers obtain a better idea of the reaction pathways/regimes under each*

*condition.*

**Response:**

OH was formed via HONO photolysis, which was produced from a photolytic process on the Teflon chamber wall as we described in the manuscript. The details can be found in a previous study on our chamber (Rohrer et al., 2005). In the revised manuscript, we have added one more sentence to better clarify this point.

"OH was formed via HONO photolysis, which was produced from a photolytic process on the Teflon chamber wall (Rohrer et al., 2005)."

In addition, in all VOC photooxidation of our study, OH was partly contributed by the recycling reaction of HO$_2$ with NO. The reaction of HO$_2$ and RO$_2$ with NO also produces NO$_2$, whose photolysis further forms O$_3$. While the detailed mechanism of O$_3$ formation is beyond the scope of this study, we have provided a brief description in the revised manuscript.

"O$_3$ was formed in photochemical reactions since NO$_x$, even in trace amount (<~1 ppbV), was present in this study."

"In the photooxidation of VOC, OH and O$_3$ often co-exist and both contribute to VOC oxidation because O$_3$ formation in chamber studies is often unavoidable during photochemical reactions of

VOC even in the presence of trace amount of $NO_x$."

In the revised manuscript, we have provided time profiles of VOC, $O_3$, OH, NO, and $NO_2$ in low $NO_x$

and high $NO_x$ conditions for α-pinene and limonene (Fig. S5). Time profiles of these species at high

$SO_2$ were similar to those at low $SO_2$ because $SO_2$ had little effect on gas phase chemistry due to its low reactivity of $SO_2$ with OH and $RO_2$, and thus are not further shown. $SO_2$ time series are shown separately (Fig. S2).

*5. Line 163. Was all the VOC reacted in each experiment?*

**Response:**

All the VOC precursor was consumed in the experiments of this study except for one experiment where small amount of VOC (~10%) was still left at the end of the reaction due to the cloudy condition and thus lower photolysis rates.

*6. Lin 163. There is no "typical" experiment in this study, as each experiment was conducted under a*

*different NOx/SO2 condition. Please clearly state that this is only for low NOx condition. Also, what*

*about high NOx condition? Was it exclusively OH reaction? Please also specify clearly.*

**Response:**

The relative dominance of OH oxidation over ozonolysis (as shown in Fig. 6) is similar in both the low $NO_x$ and high $NO_x$ conditions. At high $NO_x$, OH was often higher and meanwhile more $O_3$ was also produced.

In the revised manuscript, we have modified this sentence as follows.

"For all the experiment in this study, the VOC loss was dominated by OH oxidation over ozonolysis (see Fig. S6 as an example). The relative importance of the reaction of OH and $O_3$ with monoterpenes was similar in the low $NO_x$ and high $NO_x$ experiments. At high $NO_x$, OH was often higher while more

$O_3$ was also produced."

*7. Line 173 – 177. Here, under low NOx condition, RO2+NO dominates throughout the entire*

*experiment (RO2+HO2 only contributes to 40% at most).*

*a. These sentences clearly demonstrate the shortcomings of classifying the experiments as low NOx vs.*

*high NOx as discussed in Wennberg et al. (IGAC news, 2103). I suggest the authors to characterize*

*reactions conditions by explicitly stating the RO2 fates, rather than as low vs. high NOx.*

**Response:**

We had clearly defined our low $NO_x$ and high $NO_x$ conditions using the $RO_2$ fate in our study.

Therefore, we respectfully do not think that using the terms "low $NO_x$" and "high $NO_x$" caused ambiguity as long as we define them clearly.

In the revised manuscript, we have added the following sentence to emphasize the $RO_2$ fates under different $NO_x$ conditions.

"Note that the $RO_2$ fate in the low and high $NO_x$ conditions quantified here are further used in the discussion below since the information of $RO_2$ fate is important for data interpretation of experiments conducted at different $NO_x$ levels (Wennberg, 2013)."

*b. It is stated that under low NOx conditions, in the beginning of the experiment, a trace amount of*

*NO is formed from photolysis of HONO from the chamber wall. Is this just in the beginning of the*

*experiment, or there is a continuous NO source from HONO photolysis throughout the entire*

*experiment? Please specify.*

**Response:**

There is a continuous NO source from HONO photolysis. In the revised manuscript, we have clarified this as follows.

"The trace amount of NO (up to ~0.2 ppbV) was from the photolysis of HONO, which was continuously produced from a photolytic process on chamber walls throughout an experiment (Rohrer et al., 2005)."

*8. Line 189. The authors attributed the lower particle number concentration and nucleation rate at*

*high NOx to the decreasing RO2+RO2 reaction in the presence of NOx. However, in line 182, the*

*authors noted that RO2+RO2 reaction is negligible in this study to start with. Please reconcile these*

*seemingly contradictory statements. Also, can be suppressed nucleation under high NOx due to the*

*higher volatility of organic nitrates as compared to peroxides (from RO2+HO2)?*

**Response:**

The compounds responsible for nucleation only account for a very small fraction of $RO_2$ reaction products. Although the contribution of $RO_2+RO_2$ reaction to the total $RO_2$ loss is negligible, it can contribute a lot to the compounds responsible for nucleation because $RO_2+RO_2$ reactions form dimers, which have high molecular weight and extremely low volatility (Ehn et al., 2014; Kirkby et al., 2016).

Generally, organic nitrates are not expected to be the main compounds responsible for nucleation since their volatility is not low enough to nucleate, nor primary organic peroxides (from

$RO_2(C_{10})+HO_2$). Therefore, although under high $NO_x$ more organic nitrates were found, organic nitrates are unlikely to be the reason for the suppressed nucleation.

*9. Line 205-206. There is nucleation (from organics) in the presence of NOx as shown in Fig. 4. In*

*this sense, "absence of nucleation" here is a bit confusing. Perhaps would be clearer to say "absence*

*of seed particles".*

**Response:**

In the revised manuscript, we have modified "absence of nucleation" as follows.

"Because $NO_x$ suppressed new particle formation, the suppression of the SOA yield could be attributed to the lack of new particles as seed and thus the lack of condensational sink, or to the decrease of condensable organic materials."

*10. Line 211 and Figure 3. The author concluded that the suppression effect of NOx on SOA yields*

*was mainly due to suppression of nucleation, i.e., to the absence of particle surface as condensation*

*sink. Many critical aspects are not discussed, making this conclusion not well-justified and*

*well-supported.*

*a. If the absence of seed particle surface area is the reason for the low yield under high NOx*

*condition (at low SO2), this will point to the importance of loss of semivolatile species via chemical*

*reactions or chamber wall loss (Kroll et al., ES&T, 2007). However, the effect of vapor wall loss in*

*not considered in this study. Zhang et al. (PNAS, 2014) first systematically investigated the effects of*

*particle surface area and vapor wall loss on SOA yields. For a-pinene photooxidation and ozonolysis*

*specifically, it was found that SOA yields are largely independent of seed surface area (McVay et al.,*

*2016, Nah et al., ACP, 2016; Nah et al., ACP, 2017). Therefore, taken all these together, it is not*

*clear how the absence of particle surface area can explain the suppressed SOA yields under high NOx*

*condition in this study.*

**Response:**

Please refer to our response to a similar comment above (Pg. 2, lines 58-66).

*b. The authors dismissed the "decrease of condensable organic materials" in high NOx conditions as*

*an explanation for the observed decrease in yield. Why? If more volatile organic nitrates are formed*

*in high NOx conditions, why can't this be an explanation for the suppressed SOA yield? For limonene*

*data (Line 213-218), the authors appeared to embrace the role of volatility of oxidation products.*

**Response:**

In our study, when new particle formation was already enhanced by added $SO_2$, the SOA yield at high

$NO_x$ was comparable to that at low $NO_x$ for $\alpha$-pinene and the difference in SOA yield between high

$NO_x$ and low $NO_x$ was much smaller (Fig. 3a). If the organic materials such as organic nitrate formed in high $NO_x$ conditions were more volatile, the SOA yield in high $NO_x$ should be low regardless of

$SO_2$ concentration unless in addition to that, $SO_2$ enhanced the SOA yield at high $NO_x$ via the influence other than surface area effect, e.g., acidity effect. Organic nitrates formed at high $NO_x$ was proposed to be more volatile (Presto et al., 2005; Kroll et al., 2006). However, many organic nitrate formed in our study is highly oxidized organic molecule (HOMs) containing multi-functional groups besides nitrate ($C_{7-10}H_{9-15}NO_{8-15}$). The compounds are expected to have low volatility and they were found to have an uptake coefficient on particles of ~1 (Pullinen et al. in preparation). A recent study also implied that organic nitrate may have low volatility (Hakkinen et al., 2012). Therefore, the suppressing effect of $NO_x$ on SOA yield was more likely due to suppressed nucleation, i.e., to lack of particle surface as condensational sinks.

In the revised manuscript, we have elaborated our discussion.

"This finding can be attributed to two possible explanations. Firstly, $NO_x$ did not significantly suppress the formation of low volatile condensable organic materials, although $NO_x$ obviously suppressed the formation of products for nucleation. Secondly, $NO_x$ did suppress the formation of low-volatility condensable organic materials via forming potentially more volatile compounds and in addition to that, the suppressed formation of condensable organic materials was compensated by the presence of SO$_2$, resulting in comparable SOA yield. Organic nitrates are a group of compounds
formed at high NO$_x$, which have been proposed to be more volatile (Presto et al., 2005; Kroll et al.,
2006). However, many organic nitrates formed by photooxidation in this study were highly oxidized
organic molecules (HOMs) containing multi-functional groups besides nitrate group
(C$_{7-10}$H$_{9-15}$NO$_{8-15}$). These compounds are expected to have low volatility and they are found to have an
uptake coefficient on particles of ~1 (Pullinen et al., in preparation).    Therefore, the suppressing
effect of NO$_x$ on SOA yield was mostly likely due to suppressed nucleation, i.e., the lack of particle
surface as condensational sink."

For limonene data, please refer to our response to the comment below ("detailed comments" 10c).

*c. Line 217. How does the different range in VOC/NOx for a-pinene and limonene experiments*
*explain the differences in yields in high SO2 conditions? Please elaborate and explain clearly.*

**Response:**
The cause of the difference between the α-pinene and limonene cases is unknown for the moment and
it would be pure speculation when discussing reasons for this difference. Therefore, in the revised
manuscript, we only state that the reason is unknown so far and as possible explanation, we note that
the average volatility of limonene oxidation products may be higher at higher NO$_x$.

In the revised manuscript, we have revised this sentence as follows.

"The cause of this difference is currently unknown. Our data of SOA yield suggest that the products
formed from limonene oxidation at high NO$_x$ seemed to have higher average volatility than that at low
NO$_x$."

*11. Line 225-237. Comparisons with previous studies. Many critical details are not considered and*
*discussed. I think the authors jumped to the conclusion on whether their study agree/disagree with*
*previous studies too quickly.*

*a. Line 225. This sentence is only true for a-pinene data in this study, but not for limonene. Please*
*state clearly.*

**Response:**
In the revised manuscript, we have revised this sentence as follows.

"Our finding that the difference in SOA yield between high NO$_x$ and low NO$_x$ conditions was highly
reduced at high SO$_2$ is also in line with the findings of some previous studies using seed aerosols
(Sarrafzadeh et al., 2016; Eddingsaas et al., 2012a)."

b. *Seed particles were generated via SO2 oxidation in this study (for high SO2 experiments).*
*Previous studies directly injected seeds into the chamber. In comparing SOA yields, the author should*
*also consider the role of gas-phase chemistry and particle phase chemistry. For instance, what about*
*the reaction of SO2 and criegee intermediates? What about the effect of particle acidity on*
*particle-phase reactions (in this study vs. previous studies)? Please discuss.*

**Response:**

In our study, the reaction of $SO_2$ with Criegee intermediates was not important to the formation of
oxidized organics and SOA formation for the following reasons. 1) The reaction of VOC with $O_3$ only
contributed to a small fraction of VOC loss in this study and thus formation of Criegee intermediates
was not significant. 2) At the water vapor concentration of this study, water may compete for Criegee
intermediates with $SO_2$ to a large extent.

Particle acidity may affect the SOA yield via acid-catalyzed reactions, as we had discussed in our
manuscript. In the revised manuscript, we have elaborated this discussion by comparing with previous
studies as follows.

"Particle acidity may also play a role in affecting the SOA yield in the experiments with high $SO_2$.
Particle acidity was found to enhance the SOA yield from α-pinene photooxidation at high $NO_x$
(Offenberg et al., 2009) and "high NO" conditions (Eddingsaas et al., 2012a). Yet, in low $NO_x$
condition, particle acidity was reported to have no significant effect on the SOA yield from α-pinene
photooxidation (Eddingsaas et al., 2012a; Han et al., 2016). According to these findings, at low $NO_x$
the enhancement of SOA yield in this study is attributed to the effect of facilitating nucleation and
providing more particle surface by $SO_2$ photooxidation. At high $NO_x$, the effect in enhancing new
particle formation by $SO_2$ photooxidation seems to be more important, although the effect of particle
acidity resulted from $SO_2$ photooxidation may also play a role."

*c. The experiments in this study were conducted in the presence of humidity but previous studies were*
*mostly conducted under dry conditions. RH can affect gas-phase and particle-phase chemistry, and*
*subsequently SOA yields.*

**Response:**

We agree with the reviewer that RH can affect gas-phase and particle-phase chemistry and thus may
also affect SOA yield. Because humidity is ubiquitous in the real atmosphere, we conducted our
experiments in the presence of humidity in order to better represent ambient conditions. In the revised
manuscript, we have emphasized this point.

"RH of this study is different from many previous studies, which often used very low RH (<10%)."

*d. The authors noted that the finding that SOA yields at high NOx is comparable to that at low NOx in*
*high SO2 conditions is in line with findings in Sarrafzadeh et al. (2016) and Eddingsaas et al. (2012a).*
*I do not think that the data in Eddingsaas et al. showed that "in presence of seed aerosol, the*
*difference in the SOA yield between low and high NOx is much reduced". SOA yield is also a function*
*of deltaMo. Considering the data in Table 1 of Eddingsaas et al., the difference in yields between low*
*and high NO experiments for nucleation is 19%, and for seeded experiments are 15% and 10%.*
*However, the difference in deltaMo for the nucleation experiments is also the largest and this will*
*play a role in the yield difference.*

**Response:**

As the reviewer noticed, the data in Eddingsaas et al. (2012) showed that in the absence of seed aerosol SOA yield at low $NO_x$ is 2.5 times higher than that at high $NO_x$, while in the presence of seed aerosol SOA yield at low $NO_x$ is only 1 and 0.6 times higher than that at high $NO_x$ for neutral seed and acid seed, respectively. Therefore, the data in Eddingsaas et al. (2012) did show that "in presence of seed aerosol, the difference in the SOA yield between low and high $NO_x$ is much reduced".

We noticed that in the data given by Eddingsaas et al. (2012) the difference in $deltaM_o$ for nucleation experiments is the largest, which plays a role in SOA yield. However, the large difference in $deltaM_o$

between high $NO_x$ and low $NO_x$ cases is because the $deltaM_o$ at high $NO_x$ is the lowest in absence of seed, much lower than $deltaM_o$ in presence of seed when other conditions are largely the same. The higher $deltaM_o$ and the smaller difference in $deltaM_o$ between low $NO_x$ and high $NO_x$ in presence of seed also originated from the seed aerosol since other conditions were kept constant. This result agrees with our finding that "in presence of seed aerosol, the difference in the SOA yield between low and high $NO_x$ is much reduced".

*e. The SOA yields in this study are much lower than previous studies, why? Consid-ering the low NOx*

*low SO2 experiment, with OH dose of 1e11 molecules cm-3 s, the yield in this study is 7%. However,*

*the corresponding yield in Eddingsaas et al. is > 30% (Figure 3 of Eddingsaas et al.).*

**Response:**

The difference in SOA yield between this study and the study by Eddingsaas et al. (2012) can be explained by several reasons. Firstly, SOA yield in this study was calculated using a density of 1 g

$cm^{-3}$ while in Eddingsaas et al. (2012) SOA yield was calculated using a density of 1.32 g $cm^{-3}$.

Secondly, reaction conditions such as VOC concentrations, $NO_x$ concentrations, and OH source and concentrations of our study at low $NO_x$ are different from those in   Eddingsaas et al. (2012). For example, in our study at low $NO_x$, $RO_2$+NO account for a large fraction of $RO_2$ loss while in

Eddingsaas et al. (2012) $RO_2$+$HO_2$ is the dominant pathway of $RO_2$ loss. These differences in reaction conditions may affect SOA yield. Thirdly, the organic aerosol concentration of this study is much lower than that in Eddingsaas et al. (2012). Fourthly, the total particle surface area may be different from Eddingsaas et al. (2012) (the data are not available to compare with), which can also affect the measured SOA yield. Also note that the exceptionally high SOA yield at lower α-pinene concentration is an "outlier" to other data in Eddingsaas et al. (2012) and could not be explained by the authors.

In the revised manuscript, we have added the following discussion regarding the comparison of SOA

yield from α-pinene photooxidation with the literature.

"The reaction conditions of this study often differ from those described in the literature (see Table

S2).

The difference in these conditions can result in both different apparent dependence on specific parameters and the varied SOA yield. For example, SOA yield from α-pinene photooxidation at low

$NO_x$ in this study appeared to be much lower than that in Eddingsaas et al. (2012a). The difference between the SOA yield in this study and some of previous studies and between the values in the
literature can be attributed to several reasons: 1) $RO_2$ fates may be different. For example, in our study
at low $NO_x$, $RO_2+NO$ account for a large fraction of $RO_2$ loss while in Eddingsaas et al. (2012a)
$RO_2+HO_2$ is the dominant pathway of $RO_2$ loss. This difference in $RO_2$ fates may affect oxidation
products distribution. 2) The organic aerosol loading of this study is much lower than that some of
previous studies (e.g., Eddingsaas et al. (2012a)) (see Fig. S9). SOA yields in this study were also
plotted versus organic aerosol loading to better compare with previous studies (Fig. S8 and S9). 3)
The total particle surface area in this study may also differ from previous studies, which may
influence the apparent SOA yield due to vapor wall loss (the total particle surface area is often not
reported in many previous studies to compare with). 4) RH of this study is different from many
previous studies, which often used very low RH (<10%). It is important to emphasize that reaction
conditions including the $NO_x$ as well as $SO_2$ concentration range and RH in this study were chosen to
be relevant to the anthropogenic-biogenic interactions in the ambient atmosphere. In addition,
difference in the organic aerosol density used in yield calculation should be taken into account. In this
study, SOA yield was derived using a density of 1 g cm$^{-3}$ to better compare with many previous
studies (e.g., (Henry et al., 2012)), while in some other studies SOA yield was derived using different
density (e.g., 1.32 g cm$^{-3}$ in Eddingsaas et al. (2012a))."
In the revised manuscript, we have also added a figure plotting SOA yield as a function of organic
aerosol mass loading (Fig. S8).
*12. Line 238 onwards, effect of SO2.*
*a. One of the proposed reasons to explain the effect of SO2 is that it induces nucleation and provides*
*more particle surface area for condensation. Again, if this is the case, it will point to the importance*
*of loss of organic vapors to chamber walls, though previous studies suggested that this process does*
*not effect SOA yields from a-pinene oxidations to a large extent. With this, it is not clear if this is*
*indeed a reason for the observed SO2 effect. Please explain.*
**Response:**
Please refer to our response to the similar comments above (Pg. 2, lines 58-66; Pg. 4, lines 119-123)
and corresponding revisions to the manuscript.
*b. Line 258. Is "counterbalance" the appropriate word? If the suppression effect of NOx is*
*counterbalanced by the enhancement effect SO2, in going from "low NOx low SO2" to "high NOx*
*high SO2" one shall not observe change in SOA yields? Also, note that the limonene data showed*
*very different trends comparing to the a-pinene data. This needs to be accurately and clearly stated.*
**Response:**
We have changed "counterbalance" to "compensated". The revised sentence is as follows.
"The presence of high $SO_2$ enhanced the SOA mass yield at high $NO_x$ conditions, which was even
comparable with the SOA yield at low $NO_x$ for α-pinene oxidation. This finding indicates that the suppressing effect of $NO_x$ on SOA mass formation was ==compensated to large extent== by the presence of $SO_2$."

As for the difference in the limonene and α-pinene data, please refer to our response to the similar comments above (Pg. 1, lines 35-43) and our corresponding revisions to the manuscript.

*13. Line 315-318. This explanation is a stretch and not well-justified. There is extensive fragmentation*

*in the AMS and so the H/C ratios of oxidation product molecules do not necessarily translate to the*

*H/C ratios measured. As shown in Chhabra et al., not all experiments conducted under low NOx*

*condition have higher H/C ratios.*

**Response:**

In our opinion, the reaction pathway of $RO_2$ at least provides a likely explanation for the difference in

H/C at different $NO_x$. Although there is extensive fragmentation in the AMS, H/C measured generally reflects the H/C ratios of the overall compounds. In addition, as we discussed, Chhabra et al. (2011)'s data also show that for α-pinene photooxidation, SOA formed at high $NO_x$ generally has lower H/C, consistent with our study. Admittedly, the $RO_2$ reaction pathway is not the only factor affecting H/C

and O/C. Other factors such as VOC identity, oxidants, and reaction mechanisms, including various functionalization, fragmentation, and oligomerization in both the gas phase and particle phase also play important roles in the chemical composition and thus H/C and O/C. We did not intend to apply our explanation here to all other reaction systems.

*14. Line 262 onwards. Did the ratio of nitrate mass concentration to organics mass change over*

*time?*

**Response:**

The ratio of nitrate mass concentration to organics gradually decreased in the beginning of the reactions (2-3 h) and then leveled off. Also, note that in the very beginning of a reaction, the data have large uncertainties due to the low concentration of nitrate and organics. In our manuscript, the average ratios were used to compare different experiments. In the revised manuscript, we have clearly described this in the captions of Fig. 4.

"==The average ratios of nitrate to organics during the reaction are shown and error bars indicate the==

==standard deviations.=="

Minor comments

*1. Line 72. Why "in contrast"?*

**Response:**

The finding here is different from those in the studies discussed before. In the revised manuscript, we have modified this sentence as follows.

"In constrast, Eddingsaas et al. (2012a) found that particle yield increases with aerosol acidity ==only== in

"high NO" condition ($NO_x$ 800 ppb, α-pinene: 20-52 ppb), but is independent of the presence of seed aerosol or aerosol acidity in both ==high $NO_2$" condition ($NO_x$ 800 ppb)"== and low $NO_x$ ($NO_x$ lower than the detection limit of the $NO_x$ analyzer)."

*2. Line 84 citation. There are more studies on OH oxidations of a-pinene and they should also be*

*cited here (for example, some of the studies cited in page 2).*

**Response:**

In the revised manuscript, we have added more studies on OH oxidation of α-pinene. However, we would like to note here that many studies on α-pinene photooxidation did not quantitatively distinguish the contributions of oxidation by OH and by $O_3$.

*3. Line 125. "mass" should be "volume"? SMPS measures volume concentration.*

**Response:**

We have changed the "mass" to "volume" in the revised manuscript.

*4. Line 126. Delete "with".*

**Response:**

Done.

*5. Line 256. Sentence not clear.*

**Response:**

This sentence has been revised as follows.

"The presence of high $SO_2$ enhanced the SOA mass yield at high $NO_x$ conditions, which was even comparable with the SOA yield at low $NO_x$ for α-pinene oxidation. This finding indicates that the suppressing effect of $NO_x$ on SOA mass formation was compensated to large extent by the presence of $SO_2$."

*6.nFigure 1 caption should specify the SO2 condition.*

**Response:**

Done.

[revised manuscript text omitted]

---

## Author Response (AR2)

**Responses to Referee # 2**

We thank the reviewer for carefully reviewing our manuscript; the comments are greatly appreciated. All the comments have been addressed. We believe that revisions based on these comment have substantially improved our manuscript. In the following please find our responses to the comments one by one and the corresponding changes made to the manuscript. The original comments are shown in italics. The revised parts of the manuscript are highlighted.

*The authors have done a thorough job with the revisions and the manuscript has greatly improved. I only have two more comments:*

*1. Role of vapor wall loss:*

*The authors added a section on "Wall loss of organic vapors", Figure S4, and some other related discussions. I was not aware that the particle surface areas in these studies are so much lower than those in previous studies. In this case, I agree with the authors' comment that SOA yields from α-pinene ozonolysis or photooxidation can depend on particle surface area, when surface area is this low.*

*However, my original question remains: what is the role of vapor wall loss in the interpretation of the data (differences in yields under different $NO_x$ and $SO_2$ conditions) and conclusions presented in this work? The authors noted in the revised manuscript "For the aerosol surface area range in most of the experiments in this study, the influence of vapor wall loss on SOA yield was relatively small (<~40% for particle surface area larger than $3\times10^{-6}$ $cm^2$ $cm^{-3}$, Fig. S4). Yet, for the experiments at high $NO_x$ and low $SO_2$ for α-pinene and limonene, the influence of vapor wall loss on SOA can be high due to the low particle surface area, especially at lower $α_p$.".*

*Based on Figure S4, the value of $f_{corr}$ is roughly 12 and 4 for limonene/high $NO_x$/low $SO_2$ and a-pinene/high $NO_x$/low $SO_2$, respectively. Please also indicate the values of $f_{corr}$ for other experiments (other combinations of $NO_x$ and $SO_2$ for limonene and α-pinene experiments) on Figure S4. Based on the above sentence in the revised manuscript, is it correct that for all other experiments, the fcorr is lower than 1.4?*

*If so, how shall one interpret the differences in SOA yields shown in Figure 3, in the context of the effect of vapor wall loss on SOA yields? The high NOx / low $SO_2$ data have the lowest yields, and these are the experiments with lower particle surface areas where yields can be drastically underestimated due to vapor wall loss. In this case, the higher SOA yields obtained in high $SO_2$ experiments could potentially be explained as a result of a smaller extent of vapor wall loss in these*

*experiments? And, if one can correct for the effect of vapor wall loss, the yields under different $NO_x$ and $SO_2$ experimental conditions might be similar?*

*Overall, more clarification is needed regarding the role of vapor wall loss on the data interpretation and conclusions in this work.*

**Response:**

We thank the reviewer the supportive remarks on our revision.

Regarding Figure S4, in the revised manuscript we have updated it by adding the values of correction factors ($f_{corr}$) of all the experiments.

For all the other experiments (with particle surface area larger than $3 \times 10^{-6}$ $cm^2$ $cm^{-3}$), the $f_{corr}$ is less than 1.4 as shown from the $f_{corr}$ values in the revised Fig. S4. In the revised manuscript, we have explicitly stated this point as follows.

"For the aerosol surface area range in most of the experiments in this study (larger than $3 \times 10^{-6}$ $cm^2$ $cm^{-3}$), $f_{corr}$ is less than 1.4 (Fig. S4) and thus the influence of vapor wall loss on SOA yield was relatively small (<~40%)."

The SOA yield in the experiments at high $NO_x$ and low $SO_2$ can be substantially underestimated due to the vapor loss. When the influence of vapor wall loss is considered, the difference between high $NO_x$ and low $NO_x$ under low $SO_2$ conditions will be much reduced. And so will be the difference between high $SO_2$ and low $SO_2$ under high $NO_x$ conditions. Under high $SO_2$ experiments or low $NO_x$ conditions, the contribution of vapor wall loss to the difference in SOA yield was minor (see the values below) since the particle surface areas were higher and comparable. Nevertheless, we did not directly correct SOA yield for vapor wall loss because the correction factor ($f_{corr}$) curve in the low surface area is very steep and has very large uncertainties (Fig. S4). In addition, $\alpha_p$ also has uncertainties and may depend on the identity of each condensable compounds. This statement has been added in the revised manuscript as follows.

"We did not directly correct SOA yield for vapor wall loss because the correction factor ($f_{corr}$) curve in the low surface area range is very steep and has very large uncertainties (Fig. S4). In addition, $\alpha_p$ also has uncertainties and may depend on the identity of each condensable compounds."

In the revised manuscript, we have added the discussion on the role of vapor wall loss in SOA yield in the "Abstract", Sect. "3.3.1 Effect of $NO_x$", Sect. "3.3.2 Effect of $SO_2$", and "Conclusion" section to further clarify it as follows.

Abstract:

"This indicates that the suppression of SOA yield by $NO_x$ was largely due to the suppressed new particle formation, leading to a lack of particle surface for the organics to condense on and thus a significant influence of vapor wall loss on SOA mass yield."

"3.3.1 Effect of $NO_x$":

"Due to the low particle surface area, the wall loss of condensable vapors in the experiment at high $NO_x$ and low $SO_2$ was large (as shown by the large $f_{corr}$ in Fig. S4) and therefore SOA mass yield was suppressed. If vapor wall loss is considered, the difference between the SOA yield at high $NO_x$ and at low $NO_x$ under low $SO_2$ conditions will be much reduced, as we found for high $SO_2$ cases (Fig. 3a). Under high $SO_2$ conditions, the influence of vapor wall loss on the difference in SOA yield between high $NO_x$ and low $NO_x$ was minor (1%-8%, Fig. S4) due to the larger particle surface area."

"3.3.2 Effect of $SO_2$"

"As mentioned above, the SOA yield at high $NO_x$ and low $SO_2$ was significantly suppressed due to vapor wall loss. If the influence of vapor wall loss is considered, the SOA yield at high $NO_x$ and low $SO_2$ will be much higher and thus the observed enhancement of SOA yield by $SO_2$ under high $NO_x$ conditions will be much less pronounced. Under low $NO_x$ conditions, the influence of vapor wall loss on the difference in SOA yield between high $SO_2$ and low $SO_2$ was minor (1%-7% for α-pinene and 5-32% for limonene, see Fig. S4) due to the larger particle surface area."

"Conclusion"

"The suppression of SOA yield by $NO_x$ was likely due to the suppressed new particle formation, i.e., absence of sufficient particle surfaces for organic vapor to condense on at high $NO_x$, which could result in large vapor loss to chamber walls."

"In this study, the influence of vapor wall loss on SOA yield was estimated although the SOA yields in this study were not corrected for vapor wall loss. We need to be cautious about the enhancement of the SOA yield by $SO_2$ under high $NO_x$ conditions and the suppression of the SOA yield by $NO_x$ under low $SO_2$ conditions. These effects will be less pronounced when vapor wall loss is considered because of the significant vapor loss to chamber walls rather than to particles at low particle surface concentration. Yet, the low particle surface concentration and thus low condensational sink of vapors to particle surface reflect some real cases in the atmosphere, because when the condensational sink by particle surface is low in the atmosphere, organic vapors will be lost to the next largest sink, e.g. dry deposition. Nevertheless, our important findings hold for the influence of $NO_x$ and $SO_2$ on SOA new particle formation, mass yield, and chemical composition, showing indeed the interaction of anthropogenic and biogenic emissions in the process of SOA formation."

"In such cases, anthropogenic $NO_x$ alone may suppress the new particle formation and SOA mass from biogenic VOC oxidation, as we found in this study, because in principle the suppression of SOA mass due to suppressed nucleation can occur in the ambient atmosphere, although chamber experiments often cannot accurately simulate the vapor loss on surface in the boundary layer."

*2. Organic nitrate volatility:*

*In line 310 the authors noted that the volatility of organic nitrates is not low enough to nucleate; in line 364 it was noted that the organic nitrates are highly oxygenated and expected to have low volatility. Putting these together, are the authors suggesting that the organic nitrates are likely of low volatility but not low enough to nucleate? Please clarify this in the revised manuscript.*

**Response:**

Yes, we meant that the organic nitrates are likely of low volatility but not low enough to nucleate. In the revised manuscript, we have added the following sentences to clarify this.

[revised manuscript text omitted]

[a]: The minimum, average and maximum temperature are shown.

[b]: The average RHs of the period of monoterpenes photooxidation are shown.

Table S2. Summary of the effect of $NO_x$ on monoterpene SOA yield in the literature

| VOC | Oxidation | $NO_x$ (ppb) | Seed aerosol | OH source/OH centration (molecules $cm^{-3}$) | RH (%) | Literature |
|---|---|---|---|---|---|---|
| α-pinene 15-30 ppb; 150-200 ppb[a] | Ozonolysis | 4.6 - 2000 ppb $[VOC]_0/[NOx]_0$: 0.65-391 | No seed[b] | Low OH[c] | Dry[d] | (Presto et al., 2005) |
| α-pinene 15 ppb | Photooxidation | ≤2-1000 ppb | $(NH_4)_2SO_4$ as seed | Low $NO_x$: $3\times10^6$; high $NO_x$: initial $2\times10^7$ | 3.7-6.4 | (Ng et al., 2007) |
| α-pinene 19.8-52.4 ppb | Photooxidation | Low $NO_x$: not reported. High $NO_x$: 800 ppb | No seed, $(NH_4)_2SO_4$ or $(NH_4)_2SO_4$ +$H_2SO_4$ | Low $NO_x$: $H_2O_2$/initial OH $2\times10^6$; high $NO_x$: HONO and $CH_3NO_2$/initial OH 6-20$\times10^6$ | <10 | (Eddingsaas et al., 2012) |
| α-pinene 12 ppb | Photooxidation | 0.5 ppb-60 ppb (steady state) | No seed, $(NH_4)_2SO_4$ | $O_3$ (78 ppb)/~4-7$\times10^7$ | 63 | (Sarrafzadeh et al., 2016) |
| α-pinene 13.6-20.4 ppb | Photooxidation | Low $NO_x$: <0.3 ppb High $NO_x$: 66-82 ppb | $(NH_4)_2SO_4$ or $(NH_4)_2SO_4$ +$H_2SO_4$ | $H_2O_2$/ low $NO_x$: OH 0.8-1.1$\times10^6$; high $NO_x$: OH 4.3-5.9$\times10^6$ | 29-68 | (Han et al., 2016) |
| α-pinene 16.1-31.7 ppb | Photooxidation | Low $NO_x$: <1.9 ppb High $NO_x$: 19.675.1 ppb | $(NH_4)_2SO_4$ +$H_2SO_4$ or $H_2SO_4$+$NH_4HSO_4$ | HONO/N.A.[e] | 23-75 | (Stirnweis et al., 2017) |
| α-pinene ~20 ppb | Photooxidation | Low $NO_x$: ~0.05-0.2 ppb High $NO_x$: 20 ppb | No seed or $SO_2$ was added. | HONO/ OH: (1-15)$\times10^6$ | 28-42 | This study |

[a]: Two levels of α-pinene concentration were used.

[b]: In one high NOx, experiment $(NH_4)_2SO_4$ was used as seed aerosol.

[c]: OH scavenger was added.

[d]: RH was not specified.

[e]: OH concentrations were not specified.

[Figure]

Figure S1. Comparison of the H/C and O/C obtained using the method of Canagaratna et al. (2015) with that obtained using the method of Aiken et al. (2007).

[Figure]

Figure S2. Time series of $SO_2$ concentration in an experiment with $SO_2$ added.

[Figure]

Figure S3. Decay of $C_{10}H_{15}NO_8$ (MW 277 g/mol) in the dark chamber. Y-axis shows the natural logarithm of the peak intensity obtained from CIMS. The raw data were averaged to 1 min.

[Figure]

                                                     (a)

                                                     (b)

Figure S4. (a) Correction factor ($f_{corr}$) to account for the influence of vapor wall loss on SOA yield. The curves were derived using an average molecular weight of 200 g/mol and an accommodation coefficient ($\alpha_p$) on particles of 0.1 and 1, respectively. The lines show the $f_{corr}$ as a function of aerosol surface area concentration and solid squares show the peak aerosol surface area concentration in each experiment. The experiments corresponding to each points are shown. "α-p" and "L" denote α-pinene and limonene, respectively. "L" and "H" denote low and high, respectively. For example, "α-p-H NOx-L SO2" denote the experiment of α-pinene oxidation under high

NOx and low SO2. And the two numbers in each label box show the correction factors ($f_{corr}$) derived using $\alpha_p$ of

0.1 and 1, respectively. (b). Correction factor ($f_{corr}$) as a function of particle surface area and accommodation coefficient of organic vapors.

[Figure]

         (a)         (b)

         (c)         (d)

Figure S5. Time series of VOC, OH, NO, $NO_2$, and $O_3$ concentrations at low $SO_2$ for low $NO_x$ (a, c) and high

$NO_x$ (b, d). (a, b) and (c, d) are for limonene and α-pinene photooxidation, respectively. Similar trends were observed under high $SO_2$.

[Figure]

Figure S6. Comparison of the reaction rates of monoterpene with OH and with $O_3$ in a typical experiment of this study. The reaction rate of VOC+OH is stacked on that of VOC+$O_3$. Monoterpene oxidation was dominated by

OH oxidation. Here the data in α-pinene photooxidation at low $NO_x$ are shown. The scattering of the reaction rate of monoterpene with OH is due to the variations in the OH concentrations and OH measurement.

[Figure]

Figure S7. The concentrations of OH, HO$_2$ and RO$_2$ radicals in a typical experiment of this study. Here the data in

α-pinene photooxidation at low NO$_x$ are shown.

[Figure]

(a)

(b)

Figure S8. SOA yield from the photooxidation of α-pinene (a) and limonene (b) as a function of organic aerosol concentration ($C_{OA}$) in different $NO_x$ and $SO_2$ conditions. Both SOA yield and organic aerosol concentration were corrected for particle wall loss and dilution.

[Figure]

Figure S9. Comparison of the SOA yield as a function of organic aerosol concentration from α-pinene photooxidation at low $NO_x$ with literature data. SOA yield and organic aerosol concentration in this study were corrected for particle wall loss and dilution. SOA particle density in all studies was adjusted to 1 g cm$^{-3}$. The data for Henry et al. (2012) was extracted from Figure 2, experiment 1 in their study. The data for McVay et al. (2016)

were extracted from Figure 2-4 in their study. The data extracted from figures in the literature may be subject to uncertainties.

[Figure]

Figure S10. SOA yield at varying $SO_2$ concentrations for SOA from limonene oxidation at high $NO_x$. The $SO_2$

concentrations for low $SO_2$, moderate $SO_2$ and high $SO_2$ are <0.05 ppb, 2 ppb and 15 ppb, respectively.

[Figure]

Figure S11. The nitrogen to carbon ratio (N/C) in the SOA formed in different conditions for α-pinene and limonene oxidation. The black line, box, and whiskers show the median, 25[th] and 75[th] percentile, and 10[th] and 90[th]

percentile, respectively.

[Figure]

Figure S12. H/C and O/C ratios of SOA from photooxidation of limonene in different NO$_x$ and SO$_2$ conditions.
A: low NO$_x$, low SO$_2$, B: high NO$_x$, low SO$_2$, C: low NO$_x$, high SO$_2$, D: high NO$_x$, high SO$_2$. Note that in the
high NO$_x$, low SO$_2$ condition (panel B), the AMS signal was too low to derive reliable H/C and O/C due to the
low particle mass concentration and small particle size. Therefore, the data from an experiment with high NO$_x$
(20 ppb NO) and moderate SO$_2$ (2 ppb) is shown instead in panel B. The black dashed line corresponds to the
slope of -1.

[Figure]

Figure S13. The difference in the mass spectra of organics of the SOA from limonene photooxidation between high $NO_x$ and low $NO_x$ conditions (high $NO_x$-low $NO_x$). SOA was formed at low $SO_2$ (a) and high $SO_2$ (b). The different chemical family of high resolution mass peaks are stacked at each unit mass $m/z$ ("gt1" means greater than 1). The mass spectra were normalized to the total organic signals. Note that in the high $NO_x$, low $SO_2$ condition (panel A), the signal of AMS was too low. Therefore, the data in panel A show an experiment with high $NO_x$ (20 ppb NO) and moderate $SO_2$ (2 ppb) instead.